# Boosting of Thoughts: Trial-and-Error Problem Solving with Large Language Models

**Sijia Chen, Baochun Li**
Department of Electrical and Computer Engineering
University of Toronto
Toronto, Ontario, Canada
`sjia.chen@mail.utoronto.ca, bli@ece.toronto.edu`

**Di Niu**
Department of Electrical and Computer
Engineering, University of Alberta
Edmonton, Alberta, Canada
`dniu@ualberta.ca`

## Abstract

The reasoning performance of Large Language Models (LLMs) on a wide range of problems critically relies on chain-of-thought prompting, which involves providing a few chain of thought demonstrations as exemplars in prompts. Recent work, e.g., Tree of Thoughts, has pointed out the importance of exploration and self-evaluation in reasoning step selection for complex problem solving. In this paper, we present Boosting of Thoughts (BoT), an automated prompting framework for problem solving with LLMs by iteratively exploring and self-evaluating many trees of thoughts in order to acquire an ensemble of trial-and-error reasoning experiences, which will serve as a new form of prompting to solve the complex problem. Starting from a simple prompt without requiring examples, BoT iteratively explores and evaluates a large collection of reasoning steps, and more importantly, uses error analysis obtained from the LLM on them to explicitly revise prompting, which in turn enhances reasoning step generation, until a final answer is attained. Our experiments with GPT-4 and Llama2 across extensive complex mathematical problems demonstrate that BoT consistently achieves higher or comparable problem-solving rates than other advanced prompting approaches. The source code is available under the folder *examples/BoTReasoning* of `https://github.com/iQua/llmpebase`.

## 1 Introduction

Large language models (LLMs) with the autoregressive paradigm have gained remarkable performance across various tasks due to their potential reasoning ability Brown et al. (2020); Lewkowycz et al. (2022). The guarantee of such ability in complex tasks heavily relies on chain-of-thought (CoT) Wei et al. (2022) prompting, which provides step-by-step reasoning examples. This approach suggests that the reasoning ability can be elicited through a chain of thoughts, where a thought serves as an intermediate step toward problem solving.

Thus, subsequent studies Fu et al. (2022); Wang et al. (2022); Yao et al. (2024); Besta et al. (2023), especially Tree-of-Thought (ToT) Yao et al. (2024), have been proposed to improve CoT. To guarantee effectiveness, the prompt of these approaches generally includes human annotations on one specific task. Such a reliance limits their scalability. Recent work that either employs a double-check with LLMs to improve answers Paul et al. (2023); Weng et al. (2023); Madaan et al. (2023) or boosts prompts based on feedback Zheng et al. (2023); Zhang et al. (2023a); Hou et al. (2023); Pitis et al. (2023) has demonstrated significant promise. The existing literature generally tends to discard ineffective thoughts from the prompt. However, humans typically can continuously learn from errors by carefully analyzing them to gain experience, thereby gradually improving performance. We therefore ask: whether the thought generation of LLMs can dispense with human annotation and mimic such a problem-solving way of humans to achieve effective reasoning across various tasks?

This paper proposes a novel framework, shown in Fig. 1, referred to as the Boosting of Thoughts (BoT), which achieves the *boosting* mechanism that embraces *aggregation* and *experience*, thereby enabling the progressive refinement of unreliable reasoning steps (weak thoughts) by learning from errors to eventually solve various problems. Starting with a simple prompt without human annotations

**Figure 1:** Boosting of thoughts iteratively enhances the prompt by adding *experience*, which comprises the analysis conducted by large language models (LLM or LM) on the generated thought chain. The *experience* specifically contains the thought chain itself, the corresponding error reports, and detailed advice on revising each reasoning step. Thus, those ineffective thoughts marked with a red cross can also contribute to prompt refinement. By accumulating experiences over iterations in the prompt, BoT can eventually yield a correct thought chain starting from a simple prompt. The examples presented here are extracted from results obtained by applying GPT-4 with BoT on the `Game of 24` task.

for LLMs, BoT may get weak thoughts. With *aggregation*, BoT is capable of deriving a more logical and effective thought chain from them, thereby guiding the subsequent refinement. This guidance in our framework is achieved by tuning the prompt with *experience*, which is the detailed error reports, advice, and instructions of each reasoning step obtained by exploiting LLMs to analyze the aggregated chain. When such *experience* accumulates in the prompt, it gradually leads to stronger thoughts.

Specifically, BoT implements such a Boosting mechanism as an *experience*-driven iteration process, as shown in Fig. 1. In each iteration, for a given prompt, BoT builds massive simplistic thought structures in parallel with the LLM. We select the tree structure as in ToT Yao et al. (2024) but significantly modify it to weighted binary trees with various growth strategies for our boosting purposes. After extracting the root-to-leaf branch with the highest score per tree, the aggregation component of BoT is performed to aggregate them into one single thought chain. Subsequently, this chain is evaluated by the same LLM to gain the *experience*, which is added to the prompt as guidance for the thought generation in the next iteration.

Our contributions can be summarized in three folds. First, instead of generating more complicated structures for thoughts with well-designed prompts, this paper shows that it is possible to rely solely on a simple initial prompt, as weak thoughts can be refined progressively based on previous experience toward solving problems. Second, to achieve such a boosting mechanism, we propose Boosting of Thoughts (BoT), a novel framework that performs an *experience*-driven iterative process. Due to starting from a simple prompt, BoT is scalable across various tasks. While guaranteeing effectiveness, BoT is fast as it builds simplistic thought structures in parallel and converges to a solution after a few iterations. Finally, with GPT-4 and LlamaV2, we evaluate the performance of BoT on complex mathematical problems. Finally, relying on GPT-4 OpenAI (2023) and LlamaV2 Touvron et al. (2023), we evaluate the performance of BoT on complex mathematical problems. The problem-solving rates indicate that BoT, employing binary tree thought structures, significantly surpasses the current state-of-the-art on the `GSM8K` and `AQuA` while achieving the second-best results on other datasets. Especially on the new challenging task, `Game of 24` Yao et al. (2024), BoT is 9.7% higher than the leading approach ToT. Our BoT thus demonstrates that, through enhancing the prompt by accumulating error analysis of ineffective thought chains and the corresponding advice, even without human annotation, LLMs are scalable across various tasks while sustaining high performance.

## 2 RELATED WORK

**Multi-Step Reasoning**. The prominent work Chain-of-thought (CoT) prompting Wei et al. (2022) shows that step-by-step reasoning behaviors from LLMs can be elicited by providing intermediate reasoning steps, termed thoughts, within the prompt for each question, as also supported by Self-Consistency Wang et al. (2022) and a series of CoT-based work Zhou et al. (2023b); Fu et al. (2022). The recent work, Tree of Thoughts (ToT) Yao et al. (2024), converts the sequential reasoning process into a tree structure, in which each thought (node) may consider previous reasoning paths to produce multiple next-step thoughts. With such backtracking and expanded exploration during reasoning,

ToT performs well on problems that even challenge GPT-4 OpenAI (2023). Considering its high ability, the base thought structure of BoT largely utilizes this tree thought structure ToT. And, thanks to the boosting framework, the tree structure generated in each iteration of BoT is binary and shallow instead of the ToT's complex tree, in which each node corresponds to massive child nodes. However, the base structure is not restricted to ToT. In contrast, BoT is flexible as the base thought structure can be either ToT, GoT Besta et al. (2023), or CR Zhang et al. (2023b), where Graph of Thoughts (GoT) Besta et al. (2023) is the most recent work that expands the thought structure into a graph format. This paper will only focus on the ToT as the base thought structure and leave the usage of GoT for future work.

**Automatic Prompting**. Releasing humans from task-specific prompts attracts much attention Shin et al. (2020). To guarantee the reasoning ability of LLMs, conventional CoT Wei et al. (2022) relies on human priors to manually generate task-specific demonstrations as the prompt. However, the zero-shot CoT Kojima et al. (2022) shows that even without hand-crafted examples, by simply adding "Let's think step by step" to the prompt, LLMs are able to perform step-by-step reasoning toward accurate answers. These insights have spurred a series of subsequent studies. Auto-CoT Zhang et al. (2022) eliminates manual efforts by retrieving usable reasoning chains generated by zero-shot CoT. Active-Prompt Diao et al. (2023) first measures the uncertainty of a set of questions and thus selects only the uncertain ones to be annotated by humans. ToT Yao et al. (2024) can also reduce manual efforts, but for each task, it still requires experts to provide possible next-step thoughts in the prompt. Our paper introduces a novel boosting approach for manual-free prompting. Starting with a simple prompt, BoT iteratively enhances it based on the analysis of LLMs on thoughts.

**Prompt Engineering via Feedback**. Utilizing responses from LLMs to the input prompt as feedback for further prompt revisions has garnered much attention. Those who continuously revise the given prompt based on evaluation descriptions from LLMs aim to gain an accurate answer Weng et al. (2023). Using a similar higher-level idea of our paper, SELF-REFINE Madaan et al. (2023) proposes an iterative self-refinement algorithm to let the LLM produce feedback for its output for further refinement. PHP Zheng et al. (2023) simplifies this process by directly adding a solution from the previous answer as a hint to the subsequent prompt. REFINER Paul et al. (2023) is also related to our paper as it evaluates each reasoning step as feedback to produce a more reasonable one. Another line of research explores ensembles, particularly leveraging the boosting mechanism Freund et al. (1996) to refine the prompt using feedback from a set of examples. They adjust the prompt to focus on the unsolved problems in the previous iteration by either adding a few shot examples uncertain in the previous Pitis et al. (2023) or relying on a feedback-reflect-refine process Zhang et al. (2023a). APO Pryzant et al. (2023) iteratively refines a prompt, using the performance of the prior prompt to form a natural language for optimization. These works prove the effectiveness of the boosting mechanism in prompt engineering. However, our work is the first to highlight the importance of error analysis in enhancing the prompt toward generating effective reasoning chains. The proposed BoT extends this insight to implement an automated prompting framework by iteratively accumulating an ensemble of trial-and-error reasoning experiences in the prompt.

## 3 BOOSTING OF THOUGHTS

### 3.1 BACKGROUND

The objective of prompt engineering is to design a prompt $\mathbb{I}$ containing multiple language sequences, such that with this prompt as input, a pre-trained large language model (LLM) denoted as $p_\theta$ parameterized by $\theta$, can obtain the desired language sequence $y$. Thus, the standard Input-Output (IO) can be formulated as $y \sim p_\theta\left(y|\mathbb{I}\left(X, Q\right)\right)$ in which $\mathbb{I}\left(\cdot\right)$ means that the prompt wraps task instructions $X$ and the corresponding question $Q$.

The prompt can be designed in a more delicate way to guide the LLM toward solving a problem in a step-by-step manner. Each intermediate reasoning step is denoted as $z_i$ (a.k.a *thought*). CoT Wei et al. (2022) provides few-shot examples with the answer of each example containing a chain of *thought* $z_{1...n}$. This leads to $y \sim p_\theta\left(y|\mathbb{I}\left(\left[z_{1...n}\right]^N, X, Q\right)\right)$ where $N$ is the number of examples included in the prompt.

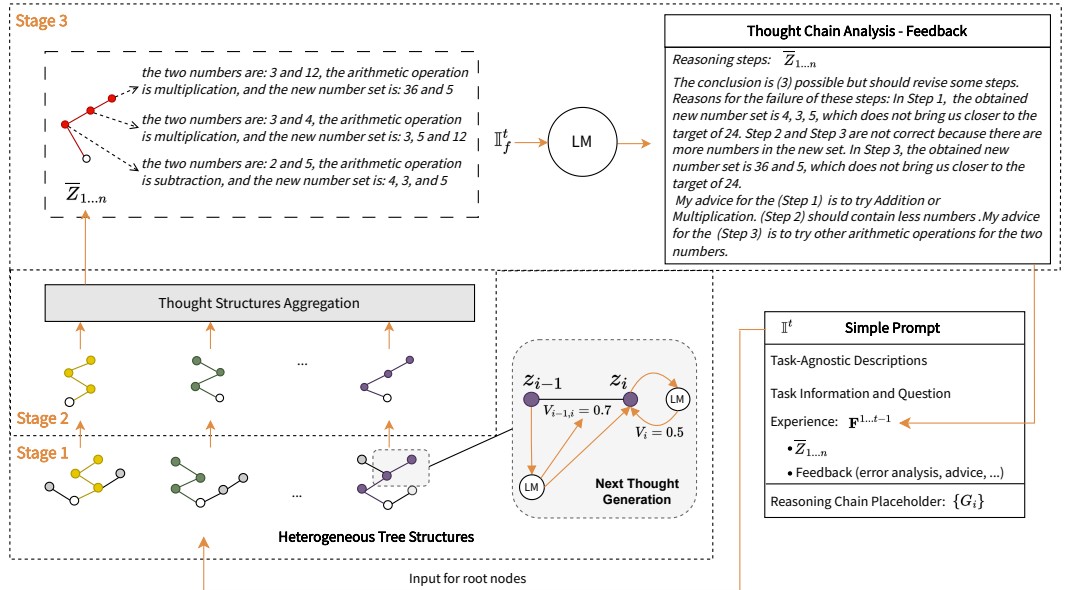

**Figure 2:** The overview of the pipeline in each iteration of BoT. To show how boosting is achieved in this *experience*-driven iteration process, we present detailed intermediate results obtained from an experiment on ChapGPT-4 on the Game of 24 dataset. Given $Q$ : "The given four numbers are: 2, 4, 5, 5", BoT performs three stages sequentially. With the simple prompt $\mathbb{I}^t$ as input, The Thought Structures Generation (Stage 1) outputs massive heterogeneous tree thought structures. Thought Structures Aggregation (Stage 2) aggregated them into a thought chain $\overline{z}_{1...n}$, which is analyzed in Stage 3 to produce *experience* to further enhance the prompt.

Instead of pre-preparing examples in the prompt, a more adaptive way is to design prompts to guide the LLMs to gradually generate the thought $z_i$ during the reasoning process. This can be formalized as $z_i \sim p_\theta \left( z_i | \mathbb{I} \left( z_{1...i-1}, X, Q \right) \right)$. Finally, the solution is formalized as $y \sim p_\theta \left( y | \mathbb{I} \left( z_{1...n}, X, Q \right) \right)$. The representative approach, ToT Yao et al. (2024), further extends this sequential reasoning steps into a tree structure in which $C$ next-step thoughts can be generated. Thus, the *thought structure* can be chain or tree.

## 3.2 FRAMEWORK

Existing literature that aims to generate the prompt with correct CoT examples or design delicate thought generation structures has three limitations. First, ineffective thoughts in those approaches are generally ignored or discarded. However, a human, who is not an expert in one field, particularly relies on analyzing previous errors to collect more experience to perform correctly on the next try. Second, they are less scalable because, for each task, an example of generating the next thoughts, such as $\mathbb{I} \left( z_1 | z_0, X, Q \right)$, should be provided in the prompt. Finally, the thought structure, such as the tree Yao et al. (2024), is generated to be overly complex to explore more reasoning steps for a better solution. This is largely due to the obtained solution may not be further revised.

In this paper, we argue that the prompt can be enhanced by continuously collecting the analysis of LLMs on those ineffective thoughts – wrong reasoning steps in a chain of thought. Therefore, even a simple prompt, such as $\mathbb{I} \left( X, Q \right)$, potentially leading to ineffective thoughts, can be progressively refined by relying on such analysis to gain powerful thoughts toward the solution.

We propose Boosting of Thoughts (BoT), an automated prompting framework incorporating, which achieves prompt *boosting* with an *experience*-driven iteration process commencing with a **simple prompt**. As summarized in Fig. 2, each iteration $t$ of BoT includes three stages. The Thought Structures Generation stage is able to fully explore reasoning chains generated by LLMs with the input prompt $\mathbb{I}^t$. In the second stage, these thought structures are aggregated to form a reasoning chain, which is to be analyzed by LLMs in the third stage to produce feedback containing error reports and detailed revision advice. Combining the aggregated reasoning chain with the feedback results in a new *experience*, denoted as $\mathbf{F}^t$. Thus, the prompt is enhanced by accumulating these experiences $\mathbf{F}^{1...t}$ over iterations.

**Simple Prompt**. For any task, in iteration $t = 0$, we create a simple initial prompt $\mathbb{I}^0\left(S, X, Q, \mathbf{F}^0, \{G_i\}\right)$, where $S$ represents task-agnostic descriptions while the terms $X$ and $Q$ respectively denote the task information and the question. The *experience* part of the prompt is denoted as $\mathbf{F}^0$, which should be empty at the beginning. $\{G_i\}$ is a placeholder that is waiting to be filled during building thought structures. In other words, when generating the next thought $z_i$, $\{G_i\}$ will be substituted with the preceding chain of thoughts $z_{1...,i-1}$.

**Thought Structures Generation**. After collecting *experience* $\mathbf{F}^{t-1}$, the prompt in the iteration $t$ can be $\mathbb{I}^t\left(S, X, Q, \mathbf{F}^{1,\cdots,t-1}, \{G_i\}\right)$. Based on this prompt, BoT generates $M$ thought structures in parallel. BoT is inherently capable of embracing any thought structure, such as the chain Wei et al. (2022) or tree Yao et al. (2024) structure. Considering the exploration of reasoning steps and experimental results, we investigate the tree thought structure. However, BoT introduces two novel modifications to make it better suited for the boosting framework.

- **Weighted Binary Tree**. With a simple prompt in each round, BoT builds the weak thoughts structured in low complexity as they can be further revised in the boosting mechanism. Thus, each thought structure of BoT is a shallow weighted binary tree. For simplicity, we retain the notation $z_{1...i-1}$ to represent the thoughts from the root to the parent of node $i$. In addition to providing each node $i$ with one thought $z_i$ and its thought evaluation score $V_i \sim p_\theta\left(z_{1...i}, \mathbb{I}_a, X, Q\right)$, we incorporate the edge score $V_{i-1,i} \sim p_\theta\left(z_{i-1}, z_i, \mathbb{I}_e, X, Q\right)$ between a child node and its parent node, where $\mathbb{I}_a$ and $\mathbb{I}_e$ refer to the instructional descriptions for thought and edge evaluations. $V_{i-1,i}$ represents the LLMs' confidence level in generating this reasoning step. Thus, the next thought generation of BoT in this tree structure is formalized as $p_\theta\left(z_i | \left(V_{i-1,i}, V_i, \mathbb{I}^t, X, Q\right)\right)$.

- **Tree Heterogeneity**. Unlike ToT Yao et al. (2024), which seeks to search for a solution in one large and complex tree, BoT aims to build highly heterogeneous tree thought structures. As a result, complete reasoning chains with various logical in trees of BoT are subsequently assessed as *experience*. Therefore, to increase heterogeneity, thought structure generation embraces different tree growth strategies, such as level-wise growth and leaf-wise growth. The former emphasizes exploration but less exploitation Chen & Guestrin (2016), while the latter does the opposite Ke et al. (2017). Thus, the leaf-wise strategy tends to continue reasoning from the current best thought to reach a better final thought as compared to level-wise growth, but it also tends to get monotonous reasoning chains. Besides, different temperature and Top_p settings of LLMs are applied. Finally, we use a small max_depth value in BoT and label a node as a leaf when its $V_{i-1,i}$ and $V_i$ values are outside the specified range $[0.3, 0.8]$.

**Thought Structures Aggregation**. Upon obtaining $M$ thought structures, BoT aggregates them into one thought chain denoted as $\overline{z}_{1...n}$. To achieve this, for each thought structure with index $m$, BoT first selects the chain with the highest evaluation score as $z_{1...n_m}^m := \arg\max_{z_{1...n} \in \mathbf{Z}^m} \sum_{i=1}^n V_i + V_{i-1,i}$ where $\mathbf{Z}^m$ denotes the set of all thought chains of $m$-th tree. Subsequently, two strategies exist to obtain $\overline{z}_{1...n}$.

- **Best-First Aggregation**. BoT relies on $\arg\max_{z_{1...n} \in \{\mathbf{Z}^m\}_{m=1}^M} \sum_{i=1}^n V_i + V_{i-1,i}$ to choose the best one as $\overline{z}_{1...n}$ from $M$ thought structures. This algorithm is fast but may lead to an unreasonable chain that is hard to guide the following refinement.

- **Greedy Aggregation**. BoT is allowed to perform a greedy search on $\{\mathbf{Z}^m\}_{m=1}^M$ to assemble a new thought chain that may not exist and is globally optimal. Starting from the initial thought, generally the root node of the tree, BoT obtains $\overline{z}_1 = \arg\max_{z_j \in \{z_1^m\}_{m=1}^M} V_j + V_{j-1,j}$. Subsequently, to obtain $\overline{z}_i$ for $\overline{z}_{i-1}$, BoT searches all thoughts where the previous step is $\overline{z}_{i-1}$ in $\{\mathbf{Z}^m\}_{m=1}^M$.

**Thought Chain Analysis**. To gain insights into what should be adjusted to enhance the prompt to generate better thoughts, BoT utilizes the self-evaluation ability Weng et al. (2023) of LLMs to assess $\overline{z}_{1...n}$. Specifically, with the prompt $\mathbb{I}_f^t\left(\overline{z}_{1...n}, X, Q\right)$ as input, LLM outputs a feedback paragraph containing issues report of this thought chain $\overline{z}_{1...n}$ and detailed advice. This feedback will be added to $\mathbf{F}^{1,\cdots,t-1}$ as a new *experience* in thought generation, resulting $\mathbf{F}^{1,\cdots,t}$.

**Iterative Refinement**. Through boosting mechanism, $\mathbf{F}^{1,\dots,t}$ is employed to iteratively enhance the prompt, resulting in $\mathbb{I}^{t+1}\left(S, X, Q, \mathbf{F}^{1,\dots,t}, \{G_i\}\right)$ for the $(t+1)$-th iteration. As the iterations progress, $\mathbf{F}^{1,\dots,t}$ may encompass many typical, unreasonable thought chains alongside those closer to a solution, all with well-defined analysis outcomes. Therefore, even when starting with a simple prompt, BoT iteratively refines this prompt to produce the correct reasoning steps leading to the accurate solution. After $T$ iterations, we utilize the $\mathbb{I}^{t+1}$ as input prompt for the LLM to gain the final answer.

## 4 EXPERIMENTS

**Datasets**. Experiments are performed on benchmark datasets with diverse mathematical problems, including MMLU Hendrycks et al. (2021a), SVAMP Patel et al. (2021), GSM8K Cobbe et al. (2021), AQuA Ling et al. (2017) and MATH Hendrycks et al. (2021b). Besides, we include a challenging mathematical reasoning task, Game of 24 Yao et al. (2024), where the goal is to use four numbers and basic arithmetic operations (addition, subtraction, multiplication, and division) to obtain 24 in 1 equation. Thus, the solution includes 3 intermediate steps.

**Competitors**. Apart from the benchmark approach, standard Input-output (IO), the comparison approaches include Chain-of-thought (CoT) Wei et al. (2022), CoT-SC Wang et al. (2022) and Complex CoT Fu et al. (2022), in which the input prompt contains a few-shot examples (8) with human annotations. Also, BoT is also compared with related works, such as Tree of thoughts (ToT) Yao et al. (2024) with the breadth limit 5, Progressive-Hint Prompting (PHP) Zheng et al. (2023), and the state-of-the-art CSV Zhou et al. (2023a).

**Large Language Models**. We conduct experiments on the two most recent models: GPT-4 OpenAI (2023) and Llama2 Touvron et al. (2023). GPT-4 is accessed via OpenAI APIs, while the llama-2-13b-chat model is downloaded from MetaAI to perform experiments locally. To construct the heterogeneous tree thought structures, BoT randomly chooses the temperature from the range of $[0.2, 0.4, 0.6, 0.7, 0.9, 1.1, 1.5]$ and the top_p from the range of $[0.1, 0.3, 0.5, 0.7, 0.9]$.

**Settings**. If not explicitly stated, BoT, in all experiments, performs $T = 10$ iterations of running and builds $M = 15$ thought structures, each being a weighted binary tree because this tends to achieve optimal results. Besides, for those benchmark datasets, we set the depth of the tree to be 5 while the corresponding depth in Game of 24 is 3. BoT+CoT means our simple prompt includes 5 examples from CoT Wei et al. (2022). In the ablation study, when no *experience* is accumulated in BoT, 8 examples of CoT will be provided in the prompt.

**Metrics**. All experiments report the Solve Rate (%) of the task as the evaluation results. To extract target answers from the output $\overline{z}_{1\dots n}^{T}$ of BoT, we specifically set the formatted description of the answer for LLMs. For commonly used datasets, the desired answer format is "The answer is: ." For the Game of 24, we utilize "Step idx, Current set: , Selected two numbers: , Operation: , Computed new number: , Remaining numbers: , New set: ". Thus, we directly compare the ground truth with the number presented in the new set. Following ToT Yao et al. (2024), we report the Solving Rate across 100 hard games as the metric.

### 4.1 MAIN RESULTS

The primary experimental results are summarized in Table. 1 and Fig. 3, where we present insights into the overall performance of BoT. Our findings indicate that the proposed BoT with Boosting mechanism 1). obtains competitive problem-solving rates in most datasets without human annotations; 2). is capable of reaching a new state-of-the-art on GSM8K and AQuA when provided with CoT examples. However, experimental results also demonstrate that BoT heavily relies on *experience*, thus is sensitive to the ability of LLMs.

Specifically, in Table. 1, BoT, starting from a simple initial prompt and performing basic chatting, eventually obtains a GSM8K solve rate 0.1% higher than the current state-of-the-art (SOTA) CSV Zhou et al. (2023a), which heavily relies on code interpreter of GPT-4. Considering AQuA, BoT is 2.5% higher than SOTA. This demonstrates that by adding error analysis and advice to the prompt without human annotations, LLMs are able to perform well on complex reasoning. The main reason is that a simple prompt can be iteratively refined by accumulating prior *experience* towards accurate problem-solving. After including CoT examples in the prompt, BoT+CoT outperforms SOTA by

**Table 1:** Utilizing BoT with GPT-4, even without human annotations, yields a notable performance enhancement. Once the simple initial prompt of BoT contains CoT examples, the corresponding approach BoT+CoT exhibits even higher solving rates. Our framework is also evaluated against leading methods such as Model Selection Zhao et al. (2023), PHP Zheng et al. (2023), and CSV Zhou et al. (2023a), each achieving state-of-the-art (SOTA) performance on the SVAMP, AQuA, and GSM8K & MATH datasets, respectively.

| Methods | No need Human Annotation | Datasets | | | | Average |
|---|---|---|---|---|---|---|
| | | SVAMP | GSM8K | AQuA | MATH | |
| SOTA | ✗ | 93.7 | 97 | 79.9 | 84.3 | 88.7 |
| Standard | ✓ | 68.7 | 87.1 | 40.6 | 42.5 | 59.7 |
| CoT | ✗ | 77.6 | 92 | 74.0 | 48.93 | 73.1 |
| Zero-shot CoT | ✓ | 74.3 | 89.6 | 73.2 | 47.7 | 71.2 |
| Complex-CoT | ✗ | 90.5 | 94.9 | 77.5 | 50.4 | 78.3 |
| PHP Complex-CoT | ✗ | 91.9 | 95.5 | 79.9 | 53.9 | 80.3 |
| BoT | ✓ | **92.7** (↓1) | **97.1** (↑0.1) | **81.4** (↑2.5) | 62.5 (↓21.8) | 83.7 (↓7.6) |
| BoT + CoT | ✗ | **94.9** (↑1.2) | **98.7** (↑1.7) | **84.9** (↑5) | 66.3 (↓18) | 86.2 (↓2.5) |

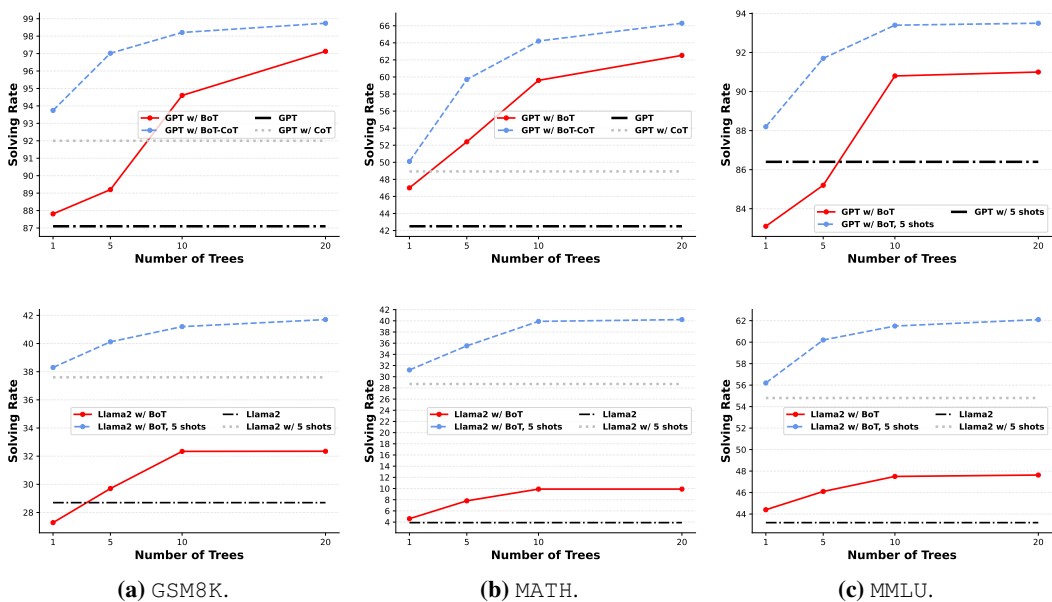

**(a)** GSM8K.  **(b)** MATH.  **(c)** MMLU.

**Figure 3:** Evaluating solve rates by applying BoT and BoT+CoT in GPT-4 OpenAI (2023) and Llama2 Touvron et al. (2023).

1.3% on average in GSM8K and AQuA datasets. We argue that the CoT examples can be regarded as the success cases in the *experience*, directly guiding the subsequent thought structures generation of BoT. Thus, cooperating with the iteration refinement, BoT+CoT reaches a new SOTA. It also deserves to show that because BoT can gradually collect analysis of various reasoning chains (bad or good) as *experience*, it is consistently close to the BoT+CoT. However, BoT and BoT+CoT, especially BoT, are at least 18% lower than SOTA in MATH. This observation means weak LLMs may not perform well with BoT due to their lower ability to analyze reasoning chains for an effective *experience*, as supported by Fig. 3.

Fig. 3 presents that with BoT, GPT-4 and Llama2 are respectively improved by 11.6% and 4.4% on average in three datasets. The two numbers show a clear trend that when the LLM is weaker, BoT's performance drops significantly. With powerful GPT-4, as presented in Fig. 3, BoT and BoT-CoT behave similarly to those shown in Table. 1. Additionally, their performance escalates along a similar trend as the number of trees varies from 1 to 20. As Llama2 is weaker, BoT is unable to benefit from its analysis to perform the *experience*-driven iteration process, which is particularly shown by Fig. 3 (a). When provided with valid success cases, i.e., 5-shots, BoT, through progressive refinement, can still help Llama2 to solve more problems than the baseline even though the improvement is limited.

## 4.2 GAME OF 24

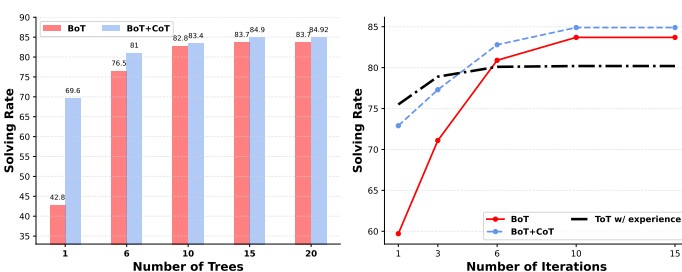

**Table 2:** Results on `Game of 24` where the settings of different approaches follow those in ToT Yao et al. (2024).

| Method | Solving rate |
|---|---|
| Standard | 7.3 |
| Standard (best of 100) | 33 |
| CoT prompt | 4 |
| CoT prompt (best of 100) | 49 |
| CoT-SC (k=100) | 9 |
| ToT | 74 |
| BoT | 83.7 |
| BoT+CoT | 84.9 |

**Figure 4:** Comparison of three approaches across varying numbers of trees and iterations.

**Table 3:** Showing aggregated thought chains and obtained *experiences* in iterations 1, 5, and 8. The given four numbers are: 2, 7, 8, 9.

| $t$-th iteration | Two numbers | Arithmetic operation | New number set | Experience | Judgement |
|---|---|---|---|---|---|
| $\mathbf{F}^1$ | 2, 8 | multiplication | 16, 7, 9 | The new set does not bring us closer to the target of 24. Try other numbers and operations. | Possible but more subsequent steps are required |
| | 9, 7 | addition | 7, 16, 16 | This step does not follow the rules of combining the remaining numbers and the obtained new number into a new set. Adjust the new set. | |
| | 16, 7 | multiplication | 16, 112 | Too many numbers in the new set. More steps are required to reach the target of 24. | |
| $\mathbf{F}^5$ | 9, 7 | addition | 16, 2, 8 | The "Evaluation Score: 0.5" is low. Increase the score. | Possible but should revise some steps |
| | 16, 8 | addition | 2, 24 | It is not possible to further manipulate the numbers to reach 24. Choose different numbers. | |
| | 2, 24 | subtraction | 22 | The new set is not correct. Can choose other two numbers. | |
| $\mathbf{F}^8$ | 9, 7 | addition | 16, 2, 8 | - | Possible |
| | 16, 2 | multiplication | 32, 8 | - | |
| | 32, 8 | subtraction | 24 | - | |

Due to the hardness of the `Game of 24` problem, GPT-4 and Llama2 both perform badly on this task, even incorporating the CoT, and CoT-SC approaches. The Llama2 model even fails to follow the correct rules of addressing the problem, making the solve rate even lower. Especially when applying BoT, which relies on the *experience*, to Llama2, all results are lower than 5% without significant improvement. Thus, we only report the performance of BoT with GPT-4. To maintain a fair comparison, we follow the settings proposed by ToT Yao et al. (2024).

As shown in Table 2, BoT without human annotations is 9.7% higher than ToT, which relies on one example showing all possible next steps. Besides, BoT+CoT, which contains 5 CoT shots in the initial prompt, is 1.2% higher than BoT. Such a close performance between BoT and BoT+CoT is attributed to the boosting mechanism, which progressively revises weak thoughts, as discussed in subsection 4.1. Adopting an *experience*-driven iterative process, BoT exhibits enhanced performance as the number of trees $M$ and the number of iterations $T$ increment. Also shown by Fig. 4, compared to BoT+CoT, BoT relies more on $M$ and $T$ as it requires to collect *experience* from a better thought chain or longer iterations. Another observation is that when enabling ToT to operate iteratively with the prompt enriched by *experience*, the problem-solving rate escalates from 72.5% in the initial iteration to 80.2% by the 10-th iteration. This demonstrates that *experience* – the analysis of previous reasoning chains can be used by LLMs to significantly improve the solve rate. However, the score obtained by ToT is still 3.5% lower than BoT. This is attributed to the fact that the aggregation stage

of BoT will produce the most representative reasoning chain in the current iteration, thus leading to more meaningful *experience* to enhance the prompt. We verify this in the ablation study section.

To better present how BoT learns from errors and previous advice, we show in Table 3 that GPT-4 is able to avoid previous errors and produce more specific advice with the increase of iteration and eventually obtain the correct solution. In the first iteration, with the simple prompt, LLMs even make a mistake in following the task rules as the new set is wrong in step 3. After analyzing, it presents correct advice on this mistake. However, the analysis at the initial iteration is vague, such as "try other numbers and operations". After five iterations, BoT aggregates multiple such analyses, deriving a more potent prompt, making the LLMs select the right numbers 9 and 7. Also, the advice is more concrete and useful. The advice for this right selection is to increase the corresponding evaluation score. Through the continuous accumulation such *experiences*, BoT progressively refines the prompt, culminating in the direct generation of a correct solution in the 8-th iteration.

## 4.3 ABLATION STUDY

**Table 4:** Comparison of BoT variations applied to GPT-4 on the `Game of 24` and `AQuA` Datasets.

| Experience $\mathbf{F}^{1\cdots t}$ | | Accumulation Type | | Game of 24 | | | AQuA | | |
|---|---|---|---|---|---|---|---|---|---|
| Issues | Advice | Replace | Add | BoT (Best first) | BoT (Greedy) | BoT (No) | BoT (Best first) | BoT (Greedy) | BoT (No) |
| ✓ | ✓ | | ✓ | 81.2 | 83.7 | 67.1 | 78 | 81.4 | 56.2 |
| | | | | 74.7 | 78.2 | 70 | 47.3 | 56.8 | 44.9 |
| ✓ | | ✓ | | 72.8 | 74.1 | 70.2 | 52.4 | 62.7 | 46.3 |
| ✓ | | | ✓ | 69.2 | 70.7 | 67.6 | 54.1 | 60 | 40.3 |
| | ✓ | ✓ | | 74.9 | 76.9 | 72.7 | 68.3 | 74.2 | 71.9 |
| | ✓ | | ✓ | 77.9 | 80 | 72.4 | 73.6 | 77 | 64.1 |

*Experience* **consistently leads to thought revision, but too much can have the opposite effect.** When the prompt accumulates issues and advice by the "adding" type, both aggregation strategies can lead to high solve rates. Maintaining a complete experience is important for revising thoughts, especially for the `AQuA` dataset, which includes wider mathematical reasoning problems. However, BoT (No), which does not perform aggregation but directly uses all reasoning chains from generated trees, suffers the worst performance in all cases, especially when the experience accumulation type is "adding". As BoT builds 15 trees each iteration, putting them all together into a prompt may cover core information, not to mention that most such experiences are invalid or harmful.

**Advice is more important to generate thoughts than others.** In all cases of Table 4, BoT variations that embrace advice as experience achieve the top solve rate. For example, with the same "adding" type, when the experience does not contain advice, the performance drops by more than $10\%$ and $20\%$ in `Game of 24` and `AQuA`, respectively. On the contrary, including issues in the *experience* serves as an auxiliary mechanism for performance improvement. Only by cooperating issues can the BoT with advice gain the best solve rate; for example, the number grows by $4.4\%$ for BoT (Greedy) in `AQuA`.

**Greedy aggregation can be the only required choice for performance purposes.** As compared to the Best-first that selects one from existing thought chains and no aggregation that maintains all thought chains, greedy aggregation adaptively merges tree structures into one better thought chain that may not exist in the current iteration. By doing so, LLM is able to perform a more meaningful analysis on a stronger thought chain, thus producing important experiences to enhance the prompt. As shown in Table 4, once the Greedy aggregation is used, BoT improves by more than $2\%$ in all cases. In `AQuA`, containing more math problems, this number is even $10\%$. Besides, as our discussion in Fig. 4, ToT with a similar experience-driven boosting mechanism reaches $80\%$ but still lags behind the BoT. This may be attributed to the inability to execute the greedy aggregation within its singular tree structure.

## 5 CONCLUSION

This paper verified that a simple prompt can be enhanced by gradually accumulating error analysis on its generated thoughts to address complex tasks. We have proposed a novel framework, the Boosting of Thoughts (BoT), to implement such progressive prompt enhancement for effective thought generation

with an *experience*-driven iteration process. Iteratively exploring and self-evaluating the generated simplistic trees of thoughts enables a simple initial prompt to be gradually enhanced by an ensemble of trial-and-error reasoning experiences, resulting in accurate solutions. Our extensive experiments demonstrated that BoT is capable of achieving state-of-the-art on multiple benchmark datasets while outperforming the alternative leading approach in Game of 24, which is a challenging mathematical reasoning task.

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

## A  Basic Prompts and Reasoning Pipeline of BoT

### A.1  Thought generation part of BoT

This subsection presents the basic prompts used by the reasoning generation of the proposed Boosting of Thoughts (BoT). For details, one can also access the source code *examples/BoostingOfThought/BoT_reasoner.py*.

**System prompt** $S$: You are an expert on mathematical problems. Perform step-by-step reasoning toward problem solving by first learning from an ensemble of trial-and-error reasoning experiences. Such trial-and-error reasoning experience specifically contains error reports and detailed advice on how to revise historical reasoning steps. Always recall these listed experiences before generating a new reasoning step, thereby avoiding making the same mistakes and reusing correct steps to generate better reasoning steps to solve the task.

**Prompt for the next thought generation**:

> f"""{task_prompt}. \n First of all, Recall historical reasoning experience: \n\n { experiences} \n\n Please make one step of reasoning to generate only one next possible reasoning step. This next reasoning step is the subsequential step from the following ordered previous steps, accompanied by their evaluated scores (A higher score means the reasoning step is more likely to complete the task.): \n\t{ chain_prompt}\n\n Based on listed previous reasoning steps (ignore them when the above space is empty), generate one single next possible step following the Task rule. (Emphasize: Please generate only one single next possible reasoning step of the given steps.)"""

where the `task_prompt` contaings the $X$ and $Q$ for the task, `experiences` is the historical reasoning experience $\mathbf{F}^{1...t}$, and `chain_prompt` is the $\{G_i\}$, which is a placeholder to be replaced by the preceding chain of thoughts $z_{1...,i-1}$ of the current thought $z_i$.

**Prompt for the thought evaluation**:

> f"""{task_prompt}. \n\n Below are the generated reasoning steps, presented in order, accompanied by their evaluated scores (A higher score means the reasoning step is more likely to complete the task.):\n{chain_prompt}\n{thought}\n\nWhat is your evaluation score for the logic, correctness, and benefit to reaching a final solution for these reasoning steps? Please select one value from [0.1, 0.3, 0.5, 0.7, 0.9, 1.0] as the score, where a higher score means better reasoning steps. The score should be placed after 'Evaluation score:' for users to read."""

where the `thought` is the current thought $z_i$.

### A.2  Experience generation part of BoT

To generate feedback for the aggregated chain, LLMs use the following basic prompts.For details, one can also access the source code *examples/BoostingOfThought/BoT_commenter.py*

**System prompt** $S$: You are an expert AI checker for math answers, dedicated to evaluating the reasoning chain generated towards addressing the mathematical problem. Judge each reasoning step of this reasoning chain by providing detailed analyses on whether the current step is a logical inference of the previous step and whether the reasoning step is beneficial to the correct solution. Provide advice and suggestions for each reasoning step with errors. Provide recommendation or rejection descriptions for each correct reasoning step.

**Prompt for the feedback**:

> f"""Given task:{task_prompt}.\n{chain_prompt}\n\n Please evaluate this reasoning chain by giving detailed comments containing the following content.\n 1.{ chain_feedback_format}. 2.{step_feedback_format}. 3.{confidence_feedback_format}.\n \n Notice: {Notice}.

where the `chain_prompt` is the aggregated thought chain $\overline{z}_{1...n}$.

**chain feedback format**: Can this reasoning chain complete the task and reach the target correctly by executing its reasoning steps? why? Write a analysis report with conclusion under 'Anlysis Report:'.

**step feedback format**: For each reasoning step, please provide a detailed analysis of whether the current step is a logical inference of the previous step and whether the reasoning step is beneficial to the correct solution. For each reasoning step with errors, please provide an error report and the corresponding advice on revision. For each reasoning step, please provide recommendation or rejection descriptions. Comments should be brief and follow the format: Reasoning step $\langle idx \rangle$. Analysis report: . Advice: . Recommendation or Reject description: .

**confidence feedback format**: What is your confidence score on these your evaluations and comments? Please select one value from [0.1, 0.3, 0.5, 0.7, 0.9, 1.0]. The score should be placed after 'Confidence score:' for users to read."

With the feedback prompt, LLMs generate reasoning experience $\mathbf{F}^t$ containing conclusion and analysis on the reasoning chain and each reasoning step.

### A.3 REASONING PIPELINE

To facilitate the understanding of the proposed Boosting of Thoughts, we summarize the reasoning pipeline in Algorithm Table 1. The source code for this pipeline can be found in the file *examples/BoostingOfThought/BoT_core.py*.

---

**Algorithm 1:** Main reasoning pipeline of BoT

**Input:** Number of iterations $T$, Number of tree structures $M$, Question $Q$.
**Output:** Aggregated chain $\overline{z}_{1...n}^T$.

1 Initialize a simple prompt $\mathbb{I}^0\left(S, X, Q, \mathbf{F}^0, \{G_i\}\right)$ where $\mathbf{F}^0$ will be an empty string.
2 **for** *each iteration* $t = 1, 2, ..., T$ **do**
3      Use LLMs with the prompt $\mathbb{I}^{t-1}\left(S, X, Q, \mathbf{F}^{t-1}, \{G_i\}\right)$ to create $M$ heterogeneous tree thought structures through Thought Structure Generation.
4      Extract thought chains $\left\{z_{i=1}^{n^m}\right\}_{m=1}^M$ from the $M$ thought structures where each $z_{i=1}^{n^m}$ is the best thought chain of $m$-th tree structure.
5      Aggregate $\left\{z_{i=1}^{n^m}\right\}_{m=1}^M$ into a single thought chain $\overline{z}_{1...n}^t$ by using either Best-First Aggregation or Greedy aggregation.
6      Perform Thought Chain Analysis on $\overline{z}_{1...n}^t$ with LLMs to obtain the feedback, which is combined with $\overline{z}_{1...n}^t$ to obtain *experience* $\mathbf{F}^t$.
7      Update the prompt by accumulating $\mathbf{F}^t$, leading to $\mathbb{I}^t\left(S, X, Q, \mathbf{F}^{t-1,t}, \{G_i\}\right)$.
8 **end**
9 Obtain the solution $\overline{z}_{1...n}^T$.

---

## B INSIGHTS FOR BOOSTING OF THOUGHTS

Boosting of Thoughts derives from our insights that the reasoning ability of large language models (LLMs) for addressing mathematical problems comes directly from experience, which contains the accumulation of the analysis and advice on previous mistakes. Once the prompt embraces valid historical reasoning experience to be recalled by LLMs before performing reasoning, the produced reasoning steps are generally more logical and reasonable, as shown in the comparison between Table 5 and 6. Such insights also made us consider that LLMs do not need to rely heavily on a well-prepared prompt with human annotations (a few chains of thought demonstrations as exemplars in prompts) for each task. Yet, as LLMs are able to learn from experience, we can start from a simple prompt without examples or manually designed content to gradually collect experience during the reasoning process. Eventually, by accumulating experiences in the prompt, LLMs achieve strong reasoning toward addressing complex problems. With these insights, the Boosting of Thoughts is designed as an automated prompting framework, which iteratively collects an ensemble of trial-and-error reasoning experiences for problem-solving with LLMs. We argue that the proposed BoT is not

an application of LLMs to specific tasks but rather builds upon the insights that LLMs' reasoning ability can be derived directly from the experience gained by analyzing incorrect reasoning chains, without relying on human priors.

To emphasize our insights, we share three key observations derived from applying gpt-3.5-turbo with a temperature of 0.7 and a top_p value of 0.7 on the `Game of 24` dataset below.

**The prompt with experience encourages LLMs to explore more logic in the responses**. As shown in Table 5, when no experience is included in the prompt, the model generates the same reasoning step five times. This observation shows one of the common problems of LLMs, which is the lack of self-motivation to explore different reasoning logics. Thus, despite their strong potential for reasoning, LLMs may become trapped in a cycle of beginning with the simplest reasoning step, which may never culminate in finding the final solution. The 'Obtained reasoning chain' part of Table 5 presents the wrong reasoning chain. We believe that as Tree of Thoughts Yao et al. (2024) generates multiple responses as thoughts to build the nodes of the Tree, such a duplicated reasoning step may lead to the failure of this algorithm in some cases. However, in the second iteration of BoT, the introduction of experience into the prompt leads to the generation of distinct initial reasoning steps, as illustrated in Table 6. The final reasoning chain can ultimately arrive at the correct solution by commencing with a wider range of potential logic. Weng et al. (2023)

**LLMs avoid making similar mistakes emphasized by the experience of the prompt**. Without including experience, which contains the error analysis, in the prompt, LLMs will make many mistakes, such as deviations from the task's rules and regressing to the initial reasoning step in the final process, as shown by Table 5's 'Obtained reasoning chain' part. After analyzing this reasoning chain and incorporating the feedback as the experience into the prompt for the second iteration of BoT, it becomes evident from Table 6 that LLMs will fully learn from the experience before engaging in reasoning. First, none of the responses replicate the same erroneous reasoning step, as illustrated by the diverse initial reasoning steps in the 'Five responses from the gpt-3.5-turbo'. Second, LLMs successfully circumvent all previously identified mistakes by rigorously adhering to task rules, eliminating incorrect reasoning, and executing logical reasoning steps. Third, it eventually leads to the correct solution for the '1 1 4 6' `Game of 24` task. Other work, such as Weng et al. (2023); Madaan et al. (2023); Zheng et al. (2023), also highlighted the importance of enhancing the prompt with the feedback, which is self-evaluation of previous answers. Nevertheless, BoT is the pioneering work founded on the insight that embracing error analysis for learning empowers LLMs to attain formidable reasoning capabilities.

**Without human annotations, LLMs automatically organize an effective reasoning chain toward solving complex problems based on experience containing error analysis.**. BoT is the pioneering initiative that proposes an automated prompting framework, leveraging the insight that LLMs can acquire effective reasoning skills for problem-solving solely through error analysis and guidance, all without the need for human annotations. As shown in Table 5, which shows the first iteration of BoT, the initial prompt only contains the basic task guidance and the question without any in-context learning examples like what in CoT. Even though the reasoning chain obtained by LLMs with such a prompt contains numerous errors and invalid reasoning steps, its error analysis and advice can be included as the experience in the input prompt to benefit the reasoning for the second iteration, as shown in Table 6. It can be observed that with no prior human guidance on correct reasoning procedures, LLMs can acquire knowledge from experience that includes error analysis and guidance from previously generated reasoning chains, leading to a progressive improvement in reasoning for problem-solving.

Therefore, our BoT provides long-term guidance for research as it demonstrates the importance of recalling error analysis and advice when enabling LLMs to generate effective reasoning processes for complex tasks. With these insights, the research of prompt engineering on inducing the reasoning ability of LLMs can focus on how to generate experience instead of introducing more human priors.

## C  THOUGHT STRUCTURES GENERATION

BoT is an automated prompting framework that iteratively accumulates the experience derived from the analysis of reasoning chains. Consequently, BoT is generalized to various thought generation methods and LLMs capable of generating and evaluating reasoning steps. And the performance of

BoT depends on the effectiveness of its thought generation structure. Thus, BoT utilizes the tree of thoughts (ToT) Yao et al. (2024), the most recent structure, as its base model to generate reasoning chains in each iteration. As mentioned in the main paper, the base thought generation model can also be the Graph of Thoughts (GoT) Besta et al. (2023), i.e., BoT with GoT. However, due to time constraints and the fact that the current GoT has not been applied to mathematical problems, BoT design exclusively embraces ToT. Furthermore, when employed as the base model within a boosting mechanism, thought structures in each iteration can remain lightweight. Ultimately, the proposed BoT generates heterogeneous tree structures, with each tree being a shallow-weighted binary tree.

## C.1 NEXT THOUGHT GENERATION AND EDGE WEIGHTS COMPUTATION

Utilizing the *Prompt for the next thought generation* discussed in Section A, LLMs can generate the next possible thought for the reasoning step $z_i$ by incorporating the experience $\mathbf{F}^1...t$ and replacing $\{G_i\}$ with $z_{1..,i-1}$. For a reasoning step $z_i$, LLMs utilize *Prompt for the thought evaluation* to generate the evaluation score as the edge weight between $z_i$ and $z_{i-1}$. For a detailed procedure, the source code is available in *examples/BoostingOfThought/BoT_reasoner.py*. In a direct example of BoT applied to the '3 5 6 8' in the `Game of 24`, using gpt-3.5-turbo, Table 7 and Table 8 present the thought generation while the Table 9 show how to compute the computation.

## C.2 THE NECESSITY OF HETEROGENEOUS TREE STRUCTURES

In each iteration BoT, the heterogeneous tree structures with different tree growth strategies and various temperature and Top_p settings of LLMs are to be built to explore more reasoning search space and improve the robustness. As can be seen in the source code *BoostingOfThought/BoT_core.py* under the *examples/*, the temperature and Top_p values for LLMs within each tree are chosen from the ranges $[0.2, 0.4, 0.6, 0.7, 0.9, 1.1, 1.5]$ and $[0.1, 0.3, 0.5, 0.7, 0.9]$, respectively. And the tree growth strategy can either be level-wise growth or leaf-wise growth. We observed the following two benefits of guaranteeing such Heterogeneity.

**Heterogeneity extends the reasoning search space, thus increasing the convergence speed**. When different trees are constructed for distinct purposes, such as exploration with a level-wise strategy or exploitation with a leaf-wise strategy, and are based on LLMs with varying configurations for being random or deterministic, the generation of reasoning steps and the resulting reasoning chains can exhibit significant differences, effectively covering a wider range of reasoning possibilities. For example, in one iteration, when LLMs generate the next thought with more confidence, similar thoughts will be explored continuously; otherwise, LLMs with more randomness tend to generate diverse thoughts. It is generally challenging to predict whether deterministic reasoning or randomness can contribute to the solution. Therefore, incorporating heterogeneity by mixing different types and logical reasoning steps allows us to comprehensively explore the reasoning space within a single iteration, ultimately facilitating subsequent iterations. In the ablation study, we compare the performance of BoT between Heterogeneous and homogeneous tree structures.

**Heterogeneity reduces the possibility of producing invalid or wrong reasoning chains, thus enhancing the robustness**. Unlike heterogeneity, in trees with homogeneous settings, individual trees tend to generate thoughts following consistent logic and build reasoning chains with the same tree structures. Then, when the logic is wrong or the underlying structure is invalid for the current question, reasoning chains obtained by all trees of BoT in each iteration can only contain noisy and incorrect reasoning steps. Even after aggregating them to obtain a more refined reasoning chain for evaluation in BoT, the experience may still diverge significantly from providing suitable problem-solving suggestions. Therefore, designing tree thought structures to be heterogeneous can be a way to reduce the possibility that there are no effective reasoning chains to be evaluated for subsequent BoT's iteration. Therefore, designing tree thought structures to be heterogeneous can help mitigate the possibility of having no effective reasoning chains available for evaluation in subsequent BoT iterations. This enhancement of robustness allows BoT to tackle questions of varying difficulty levels.

---

**Algorithm 2:** Best-First Aggregation and Greedy aggregation

---

**Input:** $M$ reasoning chains where the reasoning steps of $m$-th chain are denoted as $z_{i=1}^{n^m}$.
**Output:** Aggregated chain $\overline{z}_{1...n}$.

1   - **Best-First Aggregation**
2   **for** *each chain $m = 1, 2, ..., M$* **do**
3      Compute the sum of edge weights for $m$-th chain as $V^m = \sum_{i=m}^{n^m} V_{i-1,i}$.
4   **end**
5   Get the best chain among $M$ chains by performing $m^* = argmax_m \{V^m\}$
6   Assign the aggregated chain as the best chain, $\overline{z}_{1...n} := \left\{ z_{i=1}^{n^{m^*}} \right\}$
7   - **Greedy Aggregation**
8   $\overline{z}_1 := z_1^{m^*}$ where $m^* = argmax_m \{V_1^m\}$.
9   **for** *each aggregation step $i = 2, ..., n$* **do**
10      **for** *each chain $m = 1, 2, ..., M$* **do**
11          Collect $J^m = \left\{ j, sim\left(\overline{z}_{i-1}, z_j^m\right) > 0.7; j \in n^m \right\}$.
12          Get $j^{*,m} = argmax_{j \in J^m} \left\{ V_{j,j+1}^m \right\}$
13      **end**
14      Get the best next reasoning step by performing: $\overline{z}_i = z_{j*+1}^m$ where
         $j* = argmax_{j \in \{j^{*,m}\}_{m=1}^M} \left\{ V_{j,j+1}^m \right\}$.
15   **end**
16   Obtain the aggregated chain $\overline{z}_{1...n}$.

---

## D    THOUGHT STRUCTURES AGGREGATION

After completing the reasoning in Heterogeneous Tree Structures, the aggregation process of BoT first extracts the best reasoning chain from each tree and then combines them using either the Best-First or Greedy aggregation method into a single reasoning chain. More details of these two aggregation methods can be accessed in the source code *examples/BoostingOfThought/BoT_aggregator.py*.

As shown in the first block of the algorithm 16, the Best-first aggregation is a straightforward approach for aggregation as it directly extracts the chain with the highest sum of edge weights. This method is fast and stable. It typically guarantees competitive performance as the subsequent experience is able to be generated by analyzing the best chain among obtained reasoning chains. However, it can only select existing chains without making effective adjustments. Greedy aggregation is more advanced as it combines reasoning steps from different chains to produce a new, better reasoning chain with the highest edge weights. The greedy aggregation procedure in algorithm 16 contains two steps. It first collects reasoning steps that are similar to the aggregated reasoning step $z_{i-1}$. Thus, the next aggregated reasoning step is selected from the next reasoning steps of this collected set by maximizing the edge weights. And, $sim$ is the similarity function that uses LLMs to assess the percentage of identical words and mathematical numbers shared between two paragraphs. $0.7$ is an empirical threshold obtained from experiments.

## E    INFLUENCE OF THE BAD FEEDBACK

The feedback obtained by evaluating the aggregated reasoning chain with LLMs may include analysis of limited usefulness and completely incorrect conclusions and error reports. This issue typically arises due to the nature of LLMs, which are language models and do not inherently verify the accuracy of the generated text. Additionally, the capabilities of LLMs, such as gpt-3.5-turbo, are constrained when used as validators for mathematical problems.

A direct example is presented in Table 7. The analysis report concludes that "The final result obtained in Step 3 is 80, which is mathematically equal to 24." Even worse, the experience further contains that "the reasoning chain is correct" and "No errors were found in the reasoning steps.". Using the prompt with this experience as the input in the first iteration, BoT is misled to generate wrong reasoning steps, and the corresponding aggregated chain can be seen at the beginning of Table 8. It is evident

that the aggregated chain is logically incorrect and does not adhere to any of the rules of the `Game of 24`.

However, we argue that spurious feedback will not be amplified over iterations; instead, thanks to the iterative mechanism of BoT, its negative impact on the generated reasoning steps can be mitigated or even entirely rectified in subsequent iterations. The main reason is that the generated wrong reasoning steps will be further analyzed to produce new experiences to be added to the prompt. Specifically, as these reasoning steps contain obvious mistakes that are easy to identify, LLMs are prone to generating correct error analysis and providing effective advice for revisions. With this new experience included in the prompt, BoT is capable of generating correct reasoning steps. As demonstrated by the experience in Table 8, BoT produces detailed error reports and revision suggestions, resulting in a rational thought generation process illustrated in Table 7.

The advantage of BoT, which leverages iterations to mitigate the detrimental effects of erroneous feedback, is evident in Figure 4. Notably, the performance of BoT exhibits consistent enhancement with an increasing number of iterations. This underscores both the significance of accumulating experience iteratively and the capacity of subsequent experiences to rectify prior errors.

## F    MORE RESULTS ON MATH

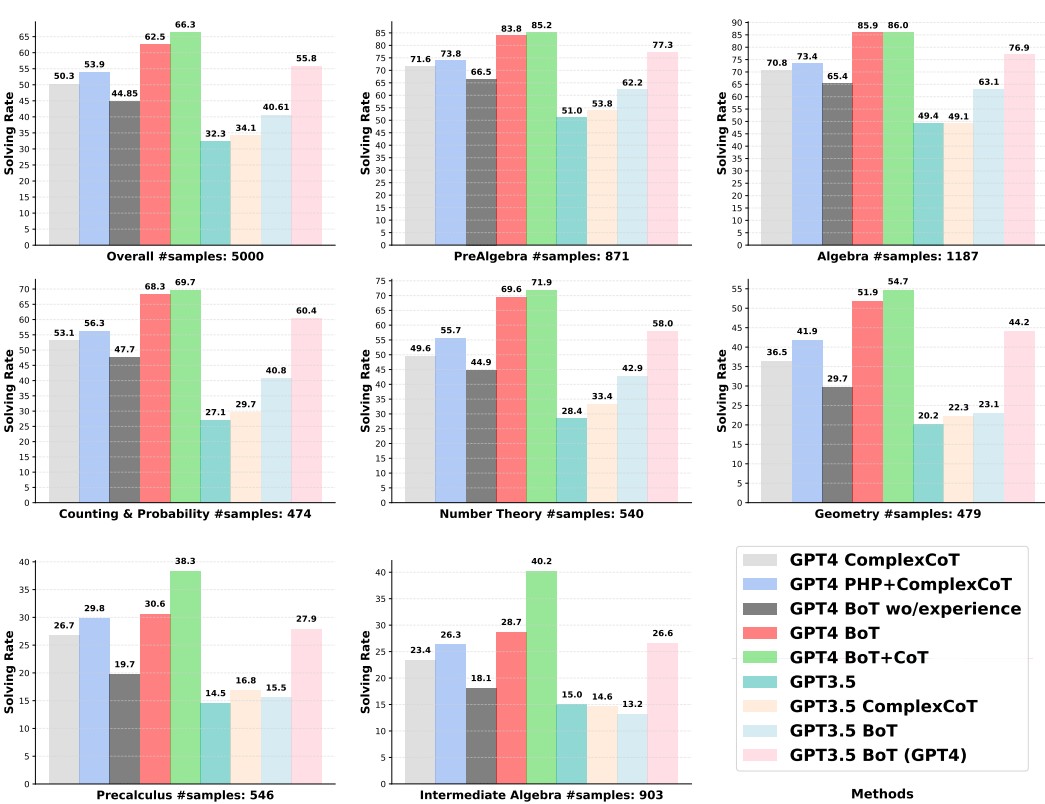

**Figure 5:** Solving rates on all the problems from different categories of the `MATH` dataset with different methods. The comparison between these methods are performed on the categories, including PreAlgebra, Algebra, Counting & Probability, Number Theory, Geometry, Precalculus, and Intermediate Algebra, of the test set. The sub-figure with the 'Overall' shows the solving rate computed on all the problems across all categories.

In Figure 5, we provide the solving rate of different methods in each category of the `MATH` dataset. The diverse range of mathematical problems in these categories poses a significantly more challenging benchmark for mathematical reasoning. Thus, the complexity and diversity of the problems in `MATH` require a wide spectrum of reasoning capabilities for solutions. Consequently, a detailed examination of our approach and its comparison with other methods in this context yields valuable insights.

**LLMs**. The experiments conducted on the `MATH` dataset employed prominent large language models (LLMs), namely, GPT-3.5-Turbo, hereafter abbreviated as GPT3.5, and GPT-4, denoted as GPT4 for brevity. We directly utilized the release APIs of OPENAI.

**Competitors**.

- GPT4 ComplexCoT. This is the GPT4 model employing greedy decoding (i.e. temperature = 0) with the ComplexCoT Fu et al. (2022) prompting method. The reasoning examples utilized in the prompt for reasoning are derived from the corresponding Complex CoT publication Fu et al. (2022). As greedy decoding is used, we do not follow the self-consistency method Wang et al. (2022) to sample reasoning paths.

- GPT3.5. With the standard prompt, the GPT3.5 model is used to generate the answer.

- GPT3.5 ComplexCoT. Similar to the GPT4 ComplexCoT but change the model to GPT3.5.

- GPT4 PHP+ComplexCoT. This is the GPT4 model employing greedy decoding (i.e. temperature = 0) with the PHP Zheng et al. (2023)+Complex CoTFu et al. (2022). Specifically, in the PHP Zheng et al. (2023) framework, the Complex CoT prompt is used to generate initial base answers, from which the PHP+Complex CoT can then develop the subsequent answer generation prompts. Thus, at the beginning of the interaction, by passing a concatenation of the base prompt of Complex CoT and the current question to the LLM, the base answer can be generated. Then, relying on the Complex CoT prompts revised into the PHP version with additional hint sentences, the progressive-hint prompting framework is performed on this base answer to update the hint over interactions to generate the right answer. We refer to this as the PHP+Complex CoT corresponding to the Progressive-Hint Prompting Complex CoT (PHP-Complex CoT) in the original work Zheng et al. (2023). The number of shots from Complex CoT is $8$.

- GPT4 BoT wo/ experience. The GPT4 model is used to perform reasoning with the proposed BoT framework without the experience accumulation. The basic settings of BoT follow those presented in the main paper. Therefore, after one iteration, the aggregated chain will be used as the solution.

- GPT4 BoT. The GPT4 is used to perform reasoning with the full version of BoT as shown in the main paper.

- GPT4 BoT + CoT. Apart from the BoT framework, $5$ reasoning examples from the CoT Wei et al. (2022) publication are included in the prompt. Therefore, in each iteration, the prompt contains not only experience but also additional $5$ CoT reasoning examples.

- GPT3.5 BoT. Similar to the GPT4 BoT but change the model to GPT3.5.

- GPT3.5 BoT (GPT4). In this experiment, we utilize the GPT3.5 to perform reasoning, thus generating thought chains in the Thought Structure Generation. However, when performing the thought evaluation and the experience generation in the aggregated Thought Chain Analysis, the GPT4 model is used to get the evaluation and the analysis feedback.

We obtain the following additional observations from the results in Figure 5.

**The top performance of BoT on challenging problems derives from the accumulation of** *experience*. BoT-related methods, such as GPT4 BoT and GPT4 BoT + CoT, consistently achieve the highest problem-solving rate on different sub-categories of `MATH`. Specifically, GPT4 BoT outperforms the current best GPT4 PHP+ComplexCoT by $8.6\%$, while GPT4 BOT + CoT is even $12.4\%$ higher. In all seven categories, GPT4 BoT is at least $0.8\%$ higher than GPT4 PHP+ComplexCoT, and the corresponding number on the Algebra problems is even $12.5\%$. Similar for GPT3.5 BoT and GPT3.5 BoT + CoT. However, when no experience is accumulated in the BoT framework, the performance drops significantly on all mathematical problems, as shown by the GPT4 BoT wo/ experience.

**In addition to experience with error analysis, including correct examples, such as simple CoT instances, is essential for improving the problem-solving efficiency of the BoT in challenging mathematical problems.**. GPT4 BoT outperforms the GPT4 PHP+ComplexCoT by a large margin on the first five sub-categories of `MATH` problems. Nevertheless, in the domains of Precalculus and Intermediate Algebra, which demand more intricate reasoning and complex logical steps for solutions, BoT exhibits only a marginal improvement of $0.8\%$ and $2.4\%$, respectively. These gains

are limited compared to the more substantial enhancements observed in simpler problem categories. After directly adding 5 correct CoT examples into the prompt, GPT-4 BoT + CoT demonstrates a significant performance boost, surpassing GPT-4 BoT by 7.7% and 11.5% in Precalculus and Intermediate Algebra domains, respectively. This basic conclusion from these observations is that to guarantee the top performance of BoT in complex mathematical problems, relying on trial-and-error analysis to learn how to reason is not sufficient; instead, the correct answers should also be provided in the prompt for LLMs.

**While GPT3.5 with BoT may initially fall behind GPT-4 CoT, leveraging GPT-4 as the evaluator and analyzer to generate experience allows GPT-3.5 BoT (GPT-4) to outperform GPT-4 Complex CoT**. With the GPT3.5, which has less capacity than GPT4, as the LLM, the solving rate obtained by BoT is at least 7.7% (on Algebra) lower than GPT4 ComplexCoT. It is evident that when less powerful LLMs produce lower-quality trial-and-error analyses, the BoT is unable to outperform GPT4 ComplexCoT. Thus, after using the GPT4 in the experience generation part while GPT3.5 is only used to generate reasoning steps, GPT3.5 BoT (GPT4) shows a significant improvement in all categories, leading to a solving rate of 55.8%, which outperforms GPT4 ComplexCoT by 5.5% and is even 1.9% higher than the current state-of-the-art GPT4 PHP+ComplexCoT. These observations further demonstrate that the accumulation of experience over iterations in the prompt constitutes the primary factor contributing to the success of the BoT framework.

## G  REASONING RESULTS OF "GAME OF 24"

First, in Table 5 - Table 9, we present the detailed prompts that BoT used during the reasoning process, thus providing a comprehensive understanding of what BoT does within each iteration. Then, starting from Table 10, we show some exact examples containing the whole reasoning process of BoT. Following the basic settings shown in the experiment section, these experiments are obtained using BoT with the GPT-3.5-turbo model.

**Table 5:** Reasoning steps generated by gpt-3.5-turbo when no experience is included in the input prompt. We first let the model generate one step of reasoning five times to check the diversity and then present the final reasoning chain after finishing the first iteration of BoT.

| Prompt of `Game of 24` without experience |
|---|
| In the game of 24, you are given four numbers, and the goal is to use basic arithmetic operations (+, -, *, /) to combine these numbers and obtain a result of 24. You can only use each number once, and parentheses can be used to change the order of operations. |
| Analysis format of each step: Step , Current set: , Selected two numbers: , Operation: , Computed new number: , Remaining numbers: , New set: . |
| The given four numbers are: 1 1 4 6. |
| Let's think step by step. |
| **Recall historical reasoning experience (Ignore when experience is empty):** |
| ######################################### |
| Pay attention to analysis and conclusion in the experience to avoid making similar mistakes by following the advice. |
| Below is a list of ordered reasoning steps, accompanied by their evaluated scores (A higher score means the reasoning step is more likely to complete the task.): |
| ——————————— |
| ——————————— |
| Based on listed reasoning steps only within the above "———" (i.e., Not the ones in the experience block), please make one step of reasoning to generate only one subsequential possible reasoning step. |
| **Five responses from the gpt-3.5-turbo** |
| Step 1, Current set: 1 1 4 6, Selected two numbers: 1 1, Operation: +, Computed new number: 2, Remaining numbers: 4 6, New set: 2 4 6., |
| Step 1, Current set: 1 1 4 6, Selected two numbers: 1 1, Operation: +, Computed new number: 2, Remaining numbers: 4 6, New set: 2 4 6., |
| Step 1, Current set: 1 1 4 6, Selected two numbers: 1 1, Operation: +, Computed new number: 2, Remaining numbers: 4 6, New set: 2 4 6., |
| Step 1, Current set: 1 1 4 6, Selected two numbers: 1 1, Operation: +, Computed new number: 2, Remaining numbers: 4 6, New set: 2 4 6., |
| Step 1, Current set: 1 1 4 6, Selected two numbers: 1 1, Operation: +, Computed new number: 2, Remaining numbers: 4 6, New set: 2 4 6., |
| **Obtained reasoning chain after aggregation** |
| Step 1, Current set: 1 1 4 6, Selected two numbers: 1 1, Operation: +, Computed new number: 2, Remaining numbers: 4 6, New set: 2 4 6.. Evaluate Score: 0.7 |
| Step 2, Current set: 1 1 4 6, Selected two numbers: 1 4, Operation: *, Computed new number: 4, Remaining numbers: 1 6, New set: 4 1 6.. Evaluate Score: 0.7 |
| Step 1, Current set: 1 1 4 6, Selected two numbers: 1 1, Operation: *, Computed new number: 1, Remaining numbers: 4 6, New set: 1 4 6. Evaluate Score: 0.7 |

## H  REASONING RESULTS OF "GSM8K"

BoT uses similar basic prompts and the specific format as shown in Table 5 - Table 9. Only the task prompt will be changed, as shown in Table 15. Then, starting from Table 16, we show some exact examples containing the whole reasoning process of BoT. Following the basic settings shown in the experiment section, these experiments are obtained using BoT with the GPT-3.5-turbo model.

**Table 6:** Reasoning steps generated by gpt-3.5-turbo when experience is included in the input prompt. This experience is the analysis on the reasoning chain of Table 5. With this experience in the prompt, we present five responses from the model and the final reasoning chain after finishing the second iteration of BoT.

| Prompt of Game of 24 with experience |
|---|
| **First of all, Recall historical reasoning experience**: |
| ######## The 1-th Reasoning Chain with Comments ######## |
| Below is a reasoning chain containing reasoning steps presented in order: |
| |
| Step 1, Current set: 1 1 4 6, Selected two numbers: 1 1, Operation: +, Computed new number: 2, Remaining numbers: 4 6, New set: 2 4 6.. Evaluate Score: 0.7 |
| Step 2, Current set: 1 1 4 6, Selected two numbers: 1 4, Operation: *, Computed new number: 4, Remaining numbers: 1 6, New set: 4 1 6.. Evaluate Score: 0.7 |
| Step 1, Current set: 1 1 4 6, Selected two numbers: 1 1, Operation: *, Computed new number: 1, Remaining numbers: 4 6, New set: 1 4 6. Evaluate Score: 0.7 |
| |
| Analysis Report: This reasoning chain cannot complete the task and reach the target correctly. The reasoning steps have errors that prevent the correct solution. |
| Reasoning Step 1: This step is a logical inference of the previous step since it follows the rule of selecting two numbers, performing an operation, and updating the set accordingly. However, the reasoning step is not beneficial to the correct solution. The computed new number of 2 does not lead to the desired result of 24. |
| Advice: Try different operations or combinations of numbers to obtain a new number that is closer to 24. |
| Recommendation or Reject description: Reject. |
| Reasoning Step 2: There is an error in the reasoning step description. It is mentioned twice as "Reasoning Step 2". However, assuming it is a unique step, it is a logical inference of the previous step. The operation of multiplication (*) is applied correctly to the selected numbers, and the new number of 4 is obtained. The remaining numbers are updated accordingly. |
| Recommendation or Reject description: Recommend. |
| Reasoning Step 3: There is an error in the reasoning step description. It is mentioned as "Reasoning Step 1" instead of "Reasoning Step 3". This step is not a logical inference of the previous step. The operation of multiplication (*) is applied to the same numbers as in Reasoning Step 1, but the computed new number of 1 is different. The remaining numbers are updated accordingly. |
| Error Report: The computation in this step is incorrect. The correct computation should be 1 * 1 = 1. |
| Advice: Double-check the computation and make sure it is accurate. |
| Recommendation or Reject description: Reject. |
| Confidence score: 0.7 |
| ######################################## |

| Five responses from the gpt-3.5-turbo |
|---|
| Step 1, Current set: 1 1 4 6, Selected two numbers: 1 4, Operation: *, Computed new number: 4, Remaining numbers: 1 6, New set: 1 4 6., |
| Step 1, Current set: 1 1 4 6, Selected two numbers: 1 6, Operation: +, Computed new number: 7, Remaining numbers: 1 4, New set: 1 4 7., |
| Step 1, Current set: 1 1 4 6, Selected two numbers: 1 1, Operation: *, Computed new number: 1, Remaining numbers: 4 6, New set: 1 4 6., |
| Step 1, Current set: 1 1 4 6, Selected two numbers: 1 4, Operation: /, Computed new number: 0.25, Remaining numbers: 1 6, New set: 0.25 1 6., |
| Step 1, Current set: 1 1 4 6, Selected two numbers: 1 1, Operation: -, Computed new number: 0, Remaining numbers: 4 6, New set: 0 4 6., |

| Obtained reasoning chain after aggregation |
|---|
| Step 1, Current set: 1 1 4 6, Selected two numbers: 1 1, Operation: *, Computed new number: 1, Remaining numbers: 4 6, New set: 1 4 6.. Evaluate Score: 0.9 |
| Step 2, Current set: 1 4 6, Selected two numbers: 1 4, Operation: *, Computed new number: 4, Remaining numbers: 6, New set: 4 6.. Evaluate Score: 0.7 |
| Step 3, Current set: 4 6, Selected two numbers: 4 6, Operation: *, Computed new number: 1, Remaining numbers: , New set: 24. Evaluate Score: 0.9 |

**Table 7:** An example of how BoT generates the next possible thought for the reasoning step $z_i$

| Prompt for the next thought generation |
|---|
| In the game of 24, you are given four numbers, and the goal is to use basic arithmetic operations (+, -, *, /) to combine these numbers and obtain a result of 24. You can only use each number once, and parentheses can be used to change the order of operations. |
| Analysis format of each step: Step , Current set: , Selected two numbers: , Operation: , Computed new number: , Remaining numbers: , New set: . |
| The given four numbers are: 3 5 6 8. |
| Let's think step by step. |
| **First of all, Recall historical reasoning experience**: |
| ######## The 1-th Reasoning Chain with Comments ######## |
| Below is a reasoning chain containing reasoning steps presented in order: |
| |
| Step 1, Current set: 3 5 6 8, Selected two numbers: 5 6, Operation: *, Computed new number: 30, Remaining numbers: 3 8, New set: 30 3 8.. Evaluate Score: 0.9 |
| Step 2, Current set: 30 3 8, Selected two numbers: 30 3, Operation: /, Computed new number: 10, Remaining numbers: 8, New set: 10 8.. Evaluate Score: 0.9 |
| Step 3, Current set: 10 8, Selected two numbers: 10 8, Operation: *, Computed new number: 80, Remaining numbers: None, New set: 80.. Evaluate Score: 0.9. |
| |
| Analysis Report: This reasoning chain can complete the task and reach the target correctly by executing its reasoning steps. The final result obtained in Step 3 is 80, which is mathematically equal to 24. |
| Reasoning Step 1: This reasoning step is a logical inference of the previous step. The numbers 5 and 6 are selected from the current set (3 5 6 8) and multiplied together to obtain the new number 30. The remaining numbers are 3 and 8. This reasoning step is beneficial to the correct solution as it reduces the number of elements in the set and generates a new number. |
| Reasoning Step 2: This reasoning step is a logical inference of the previous step. The numbers 30 and 3 are selected from the current set (30 3 8) and divided to obtain the new number 10. The remaining number is 8. This reasoning step is beneficial to the correct solution as it further reduces the number of elements in the set and generates a new number. |
| Reasoning Step 3: This reasoning step is a logical inference of the previous step. The numbers 10 and 8 are selected from the current set (10 8) and multiplied together to obtain the new number 80. Since there are no remaining numbers, the result of the reasoning chain is 80. This reasoning step is beneficial to the correct solution as it generates the final result. |
| Advice: No errors were found in the reasoning steps. |
| Recommendation or Reject Description: All reasoning steps are recommended as they are correct and contribute to reaching the correct solution. |
| Confidence score: 0.9 |
| ######## The 2-th Reasoning Chain with Comments ######## |
| **To be continue, in the Table 8** |
| ######################################## |
| Pay attention to analysis and conclusion in the experience to avoid making similar mistakes by following the advice. |
| Below is a list of ordered reasoning steps, accompanied by their evaluated scores (A higher score means the reasoning step is more likely to complete the task.): |
| |
| Step 1, Current set: 3 5 6 8, Selected two numbers: 3 8, Operation: +, Computed new number: 11, Remaining numbers: 5 6, New set: 11 5 6.. Evaluate Score: 0.7 |
| Step 2, Current set: 11 5 6, Selected two numbers: 5 6, Operation: *, Computed new number: 30, Remaining numbers: 11, New set: 30 11.. Evaluate Score: 0.9 |
| |
| Based on listed reasoning steps only within the above "————" (i.e., Not the ones in the experience block), please make one step of reasoning to generate only one subsequential possible reasoning step. |

| The generated next step |
|---|
| Step 3, Current set: 30 11, Selected two numbers: 30 11, Operation: -, Computed new number: 19, Remaining numbers: , New set: 19., |

**Table 8:** The 2-th Reasoning Chain with Comments of Table 7

######## The 2-th Reasoning Chain with Comments ########
Below is a reasoning chain containing reasoning steps presented in order:

Step 1, Current set: 10 8, Selected two numbers: 10 8, Operation: /, Computed new number: 1.25, Remaining numbers: None, New set: 1.25. Evaluate Score: 0.8.
Evaluate Score: 0.8
Step 2, Current set: 10 8, Selected two numbers: 10 8, Operation: /, Computed new number: 1.25, Remaining numbers: None, New set: 1.25.. Evaluate Score: 0.8
Step 3, Current set: 10 8, Selected two numbers: 10 8, Operation: /, Computed new number: 1.25, Remaining numbers: None, New set: 1.25.. Evaluate Score: 0.8

Analysis Report: This reasoning chain cannot complete the task and reach the target correctly. The reasoning steps provided are incorrect and do not lead to the solution of obtaining a result of 24.
Reasoning Step 1: This reasoning step is incorrect. The selected two numbers, 10 and 8, are not present in the given number set of 3, 5, 6, and 8. Additionally, the operation of division (/) does not result in the computed new number of 1.25. The remaining numbers are also not correctly identified.
Error Report: - The selected two numbers are not present in the given number set. - The operation and computed new number are incorrect. - The remaining numbers are not correctly identified.
Advice: - Ensure that the selected two numbers are present in the given number set. - Use the correct operation that will result in the desired computed new number. - Accurately identify the remaining numbers.
Recommendation or Reject Description: Reject. This reasoning step is incorrect and does not contribute to the correct solution.
Reasoning Step 2: This reasoning step is a repetition of Reasoning Step 1 and is therefore incorrect. The same errors are present in this step as in the previous step.
Error Report: - The selected two numbers are not present in the given number set. - The operation and computed new number are incorrect. - The remaining numbers are not correctly identified.
Advice: - Ensure that the selected two numbers are present in the given number set. - Use the correct operation that will result in the desired computed new number. - Accurately identify the remaining numbers.
Recommendation or Reject Description: Reject. This reasoning step is incorrect and does not contribute to the correct solution.
Reasoning Step 3: This reasoning step is a repetition of Reasoning Step 1 and Reasoning Step 2. The same errors are present in this step as in the previous steps.
Error Report: - The selected two numbers are not present in the given number set. - The operation and computed new number are incorrect. - The remaining numbers are not correctly identified.
Advice: - Ensure that the selected two numbers are present in the given number set. - Use the correct operation that will result in the desired computed new number. - Accurately identify the remaining numbers.
Recommendation or Reject Description: Reject. This reasoning step is incorrect and does not contribute to the correct solution.
Confidence Score: 0.9

**Table 9:** Computation of edge weight between $z_i$ (the step 2) and $z_{i-1}$ (the step 3)

| **Prompt for the thought evaluation** |
|---|
| In the game of 24, you are given four numbers, and the goal is to use basic arithmetic operations (+, -, *, /) to combine these numbers and obtain a result of 24. You can only use each number once, and parentheses can be used to change the order of operations. Analysis format of each step: Step , Current set: , Selected two numbers: , Operation: , Computed new number: , Remaining numbers: , New set: . The given four numbers are: 3 5 6 8. Let's think step by step. Below are the generated reasoning steps, presented in order, accompanied by their evaluated scores (A higher score means the reasoning step is more likely to complete the task.): |
| Step 1, Current set: 3 5 6 8, Selected two numbers: 3 8, Operation: +, Computed new number: 11, Remaining numbers: 5 6, New set: 11 5 6.. Evaluate Score: 0.7 Step 2, Current set: 11 5 6, Selected two numbers: 5 6, Operation: *, Computed new number: 30, Remaining numbers: 11, New set: 30 11.. Evaluate Score: 0.9 Step 3, Current set: 30 11, Selected two numbers: 30 11, Operation: -, Computed new number: 19, Remaining numbers: , New set: 19. |
| What is your evaluation score for the logic, correctness, and benefit to reaching a final solution for these reasoning steps? Please select one value from [0.1, 0.3, 0.5, 0.7, 0.9, 1.0] as the score, where a higher score means better reasoning steps. The score should be placed after 'Evaluation score:' for users to read. |
| **The generated evaluation socre** |
| Evaluation score: 0.9. |

**Table 10:** Format of prompts used in `Game of 24`.

| **Prompt Format for Next Thought Generation** |
|---|
| In the game of 24, you are given four numbers, and the goal is to use basic arithmetic operations (+, -, *, /) to combine these numbers and obtain a result of 24. You can only use each number once, and parentheses can be used to change the order of operations. Analysis format of each step: Step , Current set: , Selected two numbers: , Operation: , Computed new number: , Remaining numbers: , New set: . The given four numbers are: 5, 6, 9, 4. Let's think step by step. **Recall historical reasoning experience (Ignore when experience is empty)**: ########################################### Pay attention to analysis and conclusion in the experience to avoid making similar mistakes by following the advice. Below is a list of ordered reasoning steps, accompanied by their evaluated scores (A higher score means the reasoning step is more likely to complete the task.): |
| Based on listed reasoning steps only within the above "———" (i.e., Not the ones in the experience block), please make one step of reasoning to generate only one subsequential possible reasoning step. |
| **Prompt Format for Experience Generation** |
| Given task:In the game of 24, you are given four numbers, and the goal is to use basic arithmetic operations (+, -, *, /) to combine these numbers and obtain a result of 24. You can only use each number once, and parentheses can be used to change the order of operations. The given four numbers are: 4 5 10 11. Let's think step by step. . Below is a reasoning chain containing reasoning steps presented in order: |
| Please evaluate this reasoning chain by giving detailed comments containing the following content: 1. Can this reasoning chain complete the task and reach the target correctly by executing its reasoning steps? why? Write a analysis report with conclusion under 'Anlysis Report:'.. 2. For each reasoning step, please provide a detailed analysis of whether the current step is a logical inference of the previous step and whether the reasoning step is beneficial to the correct solution. For each reasoning step with errors, please provide an error report and the corresponding advice on revision. For each reasoning step, please provide recommendation or rejection descriptions. Comments should be brief, avoid repeating the same analysis in different steps and follow the format: Reasoning step ¡idx¿. Analysis report: . Advice: . Recommendation or Reject description: . . 3. What is your confidence score on these your evaluations and comments? Please select one value from [0.1, 0.3, 0.5, 0.7, 0.9, 1.0]. The score should be placed after 'Confidence score:' for users to read.. |

**Table 11:** BoT's success reasoning process for "4 5 10 11".

| Experience obtained in Iteration 1 |
| --- |

######## The 1-th Reasoning Chain with Comments ########
Below is a reasoning chain containing reasoning steps presented in order:

Step 1, Current set: 4 5 10 11, Selected two numbers: 4 5, Operation: *, Computed new number: 20, Remaining numbers: 10 11, New set: 20 10 11.. Evaluate Score: 0.7
Step 2, Current set: 20 10 11, Selected two numbers: 10 11, Operation: *, Computed new number: 110, Remaining numbers: 20, New set: 110 20.. Evaluate Score: 0.7
Step 3, Current set: 110 20, Selected two numbers: 110 20, Operation: /, Computed new number: 5.5, Remaining numbers: None, New set: 5.5.. Evaluate Score: 0.9

Analysis Report:
This reasoning chain can complete the task and reach the target correctly by executing its reasoning steps. The final result is 5.5, which is mathematically equal to 24.
Reasoning step 1: This step is a logical inference of the given numbers. The numbers 4 and 5 are selected from the current set, and the operation * is applied to obtain the new number 20. The remaining numbers are 10 and 11. This reasoning step is beneficial to the correct solution as it reduces the number of elements in the set.
Reasoning step 2: This step is a logical inference of the previous step. The numbers 10 and 11 are selected from the current set, and the operation * is applied to obtain the new number 110. The remaining number is 20. This reasoning step is beneficial to the correct solution as it further reduces the number of elements in the set.
Reasoning step 3: This step is a logical inference of the previous step. The numbers 110 and 20 are selected from the current set, and the operation / is applied to obtain the new number 5.5. There are no remaining numbers, which means the reasoning chain has reached the result. The final result of 5.5 is mathematically equal to 24.
Advice: No errors were found in this reasoning chain. The steps are correctly executed and lead to the correct solution.
Recommendation or Reject description: The reasoning chain is recommended as it successfully reaches the target and provides the correct solution.
Confidence score: 0.9
#########################################

| Experience obtained in Iteration 2 |
| --- |

######## The 2-th Reasoning Chain with Comments ########
Below is a reasoning chain containing reasoning steps presented in order:

Step 1, Current set: 4 5 10 11, Selected two numbers: 4 10, Operation: *, Computed new number: 40, Remaining numbers: 5 11, New set: 40 5 11.. Evaluate Score: 0.9
Step 2, Current set: 40 5 11, Selected two numbers: 5 11, Operation: *, Computed new number: 55, Remaining numbers: 40, New set: 55 40.. Evaluate Score: 0.9
Step 1, Current set: 4 5 10 11, Selected two numbers: 4 10, Operation: *, Computed new number: 40, Remaining numbers: 5 11, New set: 40 5 11.. Evaluate Score: 0.9

Anlysis Report: This reasoning chain cannot complete the task and reach the target correctly. The reasoning steps are repetitive and do not lead to the desired result of 24.
Reasoning step 1: This step is a logical inference of the given numbers and follows the rules of the game. Multiplying 4 and 10 gives 40, and the remaining numbers are 5 and 11. However, this step does not contribute to the correct solution of reaching 24.
Reasoning step 2: This step is a repetition of step 1. It selects the remaining numbers 5 and 11 and multiplies them to get 55. The new set becomes 55 and 40. Again, this step does not lead to the target of 24.
Reasoning step 3: This step is a repetition of step 1. It does not provide any new information or progress towards the target. The result is the same as in step 1.
Advice: The reasoning steps need to be revised and adjusted to find a different approach that leads to the target of 24. Repetitive steps will not help in reaching the correct solution.
Recommendation or Reject description: The reasoning chain is rejected due to repetitive steps and lack of progress towards the target.
Confidence score: 0.7
#########################################

| Experience obtained in Iteration 3 |
| --- |

######## The 3-th Reasoning Chain with Comments ########
Below is a reasoning chain containing reasoning steps presented in order:

Step 1, Current set: 4 5 10 11, Selected two numbers: 4 5, Operation: +, Computed new number: 9, Remaining numbers: 10 11, New set: 9 10 11.. Evaluate Score: 0.7
Step 2, Current set: 9 10 11, Selected two numbers: 9 10, Operation: *, Computed new number: 90, Remaining numbers: 11, New set: 90 11.. Evaluate Score: 0.7
Step 3, Current set: 90 11, Selected two numbers: 90 11, Operation: /, Computed new number: 8.181818181818182, Remaining numbers: , New set: 8.181818181818182.. Evaluate Score: 0.7

Analysis Report: This reasoning chain cannot complete the task and reach the target correctly. The final result obtained in Step 3 is 8.181818181818182, which is not equal to the target number 24. Therefore, this reasoning chain is incorrect.
Reasoning step 1: This step is a logical inference of the previous step. Two numbers, 4 and 5, are selected from the current set (4 5 10 11) and added together to obtain a new number, 9. The remaining numbers are 10 and 11, and the new set becomes 9 10 11. This step is beneficial to the correct solution as it combines two numbers and reduces the size of the set.
Recommendation or Reject description: Recommend.
Reasoning step 2: This step is a logical inference of the previous step. Two numbers, 9 and 10, are selected from the current set (9 10 11) and multiplied together to obtain a new number, 90. The remaining number is 11, and the new set becomes 90 11. This step is beneficial to the correct solution as it combines two numbers and reduces the size of the set.
Recommendation or Reject description: Recommend.
Reasoning step 3: This step is a logical inference of the previous step. Two numbers, 90 and 11, are selected from the current set (90 11) and divided to obtain a new number, 8.181818181818182. Since there are no remaining numbers, this is the final result. However, the obtained result is not equal to the target number 24.
Error report: The reasoning step is incorrect as the obtained result is not equal to the target number.
Advice: Try different operations or combinations of numbers to obtain a result closer to the target number.
Confidence score: 0.7
#########################################

| To be continue in Table 12 |
| --- |

**Table 12:** BoT's success reasoning process for "4 5 10 11".

| Experience obtained in Iteration 4 |
| --- |
| ######## The 4-th Reasoning Chain with Comments ########
Below is a reasoning chain containing reasoning steps presented in order:

Step 1, Current set: 4 5 10 11, Selected two numbers: 4 10, Operation: *, Computed new number: 40, Remaining numbers: 5 11, New set: 40 5 11.. Evaluate Score: 0.7
Step 2, Current set: 40 5 11, Selected two numbers: 40 5, Operation: +, Computed new number: 45, Remaining numbers: 11, New set: 45 11.. Evaluate Score: 0.7
Step 3, Current set: 45 11, Selected two numbers: 45 11, Operation: -, Computed new number: 34, Remaining numbers: , New set: 34.. Evaluate Score: 0.9

Analysis Report: This reasoning chain can complete the task and reach the target correctly by executing its reasoning steps. The final step computes the number 34, which is the desired result of 24 when using the given four numbers and the specified arithmetic operations.
Reasoning step 1: The current set is 4 5 10 11. Two numbers, 4 and 10, are selected. The operation chosen is multiplication (*), resulting in a computed new number of 40. The remaining numbers are 5 and 11. The new set becomes 40 5 11. Logical inference: The selected numbers are valid elements from the current set, and the operation is a valid arithmetic operation. Beneficial to the correct solution: Yes, this step reduces the number of elements in the set and computes a new number that can be used in subsequent steps. Recommendation: Accept.
Reasoning step 2: The current set is 40 5 11. Two numbers, 40 and 5, are selected. The operation chosen is addition (+), resulting in a computed new number of 45. The remaining number is 11. The new set becomes 45 11. Logical inference: The selected numbers are valid elements from the current set, and the operation is a valid arithmetic operation. Beneficial to the correct solution: Yes, this step further reduces the number of elements in the set and computes a new number that can be used in subsequent steps. Recommendation: Accept.
Reasoning step 3: The current set is 45 11. Two numbers, 45 and 11, are selected. The operation chosen is subtraction (-), resulting in a computed new number of 34. There are no remaining numbers. The new set becomes 34. Logical inference: The selected numbers are valid elements from the current set, and the operation is a valid arithmetic operation. Beneficial to the correct solution: Yes, this step computes the final result of 34, which is not equal to the target of 24. Recommendation: Accept.
Confidence score: 0.9
######################################## |
| **Experience obtained in Iteration 5** |
| ######## The 5-th Reasoning Chain with Comments ########
Below is a reasoning chain containing reasoning steps presented in order:

Step 1, Current set: 4 5 10 11, Selected two numbers: 4 10, Operation: *, Computed new number: 40, Remaining numbers: 5 11, New set: 40 5 11.. Evaluate Score: 0.9
Step 2, Current set: 40 5 11, Selected two numbers: 5 11, Operation: *, Computed new number: 55, Remaining numbers: 40, New set: 55 40.. Evaluate Score: 0.9
Step 3, Current set: 55 40, Selected two numbers: 55 40, Operation: +, Computed new number: 95, Remaining numbers: None, New set: 95.. Evaluate Score: 1.0

Analysis Report: This reasoning chain can complete the task and reach the target correctly by executing its reasoning steps. The final result obtained in Step 3 is 95, which is not equal to 24 mathematically. Therefore, this reasoning chain does not provide the correct solution.
Reasoning step 1: The current step is a logical inference of the previous step, as it follows the rule of selecting two numbers from the current set and performing the * operation. The reasoning step is beneficial to the correct solution as it generates a new number and updates the set for the next step.
Reasoning step 2: The current step is a logical inference of the previous step, as it follows the rule of selecting two numbers from the current set and performing the * operation. The reasoning step is beneficial to the correct solution as it generates a new number and updates the set for the next step. However, the reasoning step is not beneficial to the correct solution, as the final result obtained is 95, which is not equal to 24 mathematically
Reasoning step 3: The current step is a logical inference of the previous step, as it follows the rule of selecting two numbers from the current set and performing the + operation. However, the reasoning step is not beneficial to the correct solution, as the final result obtained is 95, which is not equal to 24 mathematically.
Advice: In Step 2, instead of performing the * operation, a different operation should be chosen to obtain a new number. In Step 3, instead of performing the + operation, a different operation should be chosen to obtain a new number that can be combined with the remaining number(s) to reach the target of 24.
Recommendation or Reject description: Step 1: Recommendation Step 2: Reject Step 3: Reject
Confidence score: 0.9
######################################## |
| **Experience obtained in Iteration 6** |
| ######## The 6-th Reasoning Chain with Comments ########
Below is a reasoning chain containing reasoning steps presented in order:

Step 1: Current set: 4 5 10 11, Selected two numbers: 4 10, Operation: *, Computed new number: 40, Remaining numbers: 5 11, New set: 40 5 11.. Evaluate Score: 0.9
Step 2: Current set: 40 5 11, Selected two numbers: 5 11, Operation: +, Computed new number: 16, Remaining numbers: 40, New set: 16 40.. Evaluate Score: 0.9
Step 3: Current set: 16 40, Selected two numbers: 16 40, Operation: +, Computed new number: 56, Remaining numbers: None, New set: 56. Evaluate Score: 0.9

Analysis Report: This reasoning chain can complete the task and reach the target correctly by executing its reasoning steps. The final result obtained in the last step is 56, which is not equal to the target number 24.
Step 1: This step is a logical inference of the given numbers and the goal of the game. The selected two numbers, 4 and 10, are multiplied to obtain the new number 40. The remaining numbers, 5 and 11, are correctly included in the new set. This step is beneficial to the correct solution as it reduces the number of elements in the set and progresses towards the target.
Step 2: This step is a logical inference of the previous step. The selected two numbers, 5 and 11, are added to obtain the new number 16. The remaining number, 40, is correctly included in the new set. This step is beneficial to the correct solution as it reduces the number of elements in the set and progresses towards the target.
Step 3: This step is a logical inference of the previous step. The selected two numbers, 16 and 40, are added to obtain the new number 56. Since there are no remaining numbers, the reasoning chain ends here. However, the obtained result of 56 does not match the target number of 24. Therefore, this reasoning step is incorrect.
Error Report: The reasoning step in Step 3 is incorrect because the obtained result of 56 does not match the target number of 24.
Advice: To correct the reasoning step in Step 3, a different operation or combination of numbers should be chosen to obtain a new number that is closer to the target of 24.
Recommendation or Reject description: Step 1: Recommended Step 2: Recommended Step 3: Rejected
Confidence score: 0.9
######################################## |
| **To be continue in Table 13** |

**Table 13:** BoT's success reasoning process for "4 5 10 11".

| Experience obtained in Iteration 7 |
| --- |
| ######## The 7-th Reasoning Chain with Comments ########
Below is a reasoning chain containing reasoning steps presented in order:

Step 1: Current set: 4 5 10 11, Selected two numbers: 4 10, Operation: *, Computed new number: 40, Remaining numbers: 5 11, New set: 40 5 11.. Evaluate Score: 0.9
Step 2: Current set: 40 5 11, Selected two numbers: 5 11, Operation: +, Computed new number: 16, Remaining numbers: 40, New set: 16 40.. Evaluate Score: 0.9
Step 3: Current set: 16 40, Selected two numbers: 16 40, Operation: -, Computed new number: 24, Remaining numbers: None, New set: 24. Evaluate Score: 0.9

Analysis Report: This reasoning chain can complete the task and reach the target correctly by executing its reasoning steps. The final result obtained in the last step is 24, which is mathematically equal to the target number 24.
Step 1: This step is a logical inference of the given numbers and the goal of the game. The selected two numbers, 4 and 10, are multiplied to obtain the new number 40. The remaining numbers, 5 and 11, are correctly included in the new set. This step is beneficial to the correct solution as it reduces the number of elements in the set and progresses towards the target.
Step 2: This step is a logical inference of the previous step. The selected two numbers, 5 and 11, are added to obtain the new number 16. The remaining number, 40, is correctly included in the new set. This step is beneficial to the correct solution as it reduces the number of elements in the set and progresses towards the target.
Step 3: This step is a logical inference of the previous step. The selected two numbers, 16 and 40, are substracted to obtain the new number 24. Since there are no remaining numbers, the reasoning chain ends here. The final result of 24 is mathematically equal to 24.
No errors were found in this reasoning chain. The steps are correctly executed and lead to the correct solution.
Recommendation or Reject description: The reasoning chain is recommended as it successfully reaches the target and provides the correct solution.
Confidence score: 0.9
###################################### |

| Experience obtained in Iteration 8 |
| --- |
| ######## The 8-th Reasoning Chain with Comments ########
Below is a reasoning chain containing reasoning steps presented in order:

Step 1: Current set: 4 5 10 11, Selected two numbers: 4 10, Operation: *, Computed new number: 40, Remaining numbers: 5 11, New set: 40 5 11.. Evaluate Score: 0.9
Step 2: Current set: 40 5 11, Selected two numbers: 5 11, Operation: +, Computed new number: 16, Remaining numbers: 40, New set: 16 40.. Evaluate Score: 0.9
Step 3: Current set: 16 40, Selected two numbers: 16 40, Operation: -, Computed new number: 24, Remaining numbers: None, New set: 24. Evaluate Score: 0.9

Analysis Report: This reasoning chain can successfully complete the task and reach the target correctly by executing its reasoning steps. The final result of 24 is mathematically correct and equal to 24.
Reasoning step 1: This reasoning step is a logical inference of the given task rule. It follows the rule of selecting two numbers (4 and 10) and performing the operation of multiplication to obtain a new number (40). The remaining numbers (5 and 11) are correctly updated to the new set (40 5 11). This reasoning step is beneficial to the correct solution.
Reasoning step 2: This reasoning step is a logical inference of the previous step. It follows the rule of selecting two numbers (5 and 11) and performing the operation of addition to obtain a new number (16). The remaining number (40) is correctly updated to the new set (16 40). This reasoning step is beneficial to the correct solution.
Reasoning step 3: This reasoning step is a logical inference of the previous step. It follows the rule of selecting two numbers (16 and 40) and performing the operation of subtraction to obtain a new number (24). Since there are no remaining numbers, the new set is empty. The final result of 24 is mathematically correct and equal to 24. This reasoning step is beneficial to the correct solution.
Advice: No errors were found in this reasoning chain. The steps are correctly executed according to the given task rule.
Recommendation or Reject description: All reasoning steps in this reasoning chain are recommended as they are correct and contribute to the correct solution.
Confidence score: 0.9
###################################### |

| Experience obtained in Iteration 9 |
| --- |
| ######## The 9-th Reasoning Chain with Comments ########
Below is a reasoning chain containing reasoning steps presented in order:

Step 1: Current set: 4 5 10 11, Selected two numbers: 4 10, Operation: *, Computed new number: 40, Remaining numbers: 5 11, New set: 40 5 11.. Evaluate Score: 0.9
Step 2: Current set: 40 5 11, Selected two numbers: 5 11, Operation: +, Computed new number: 16, Remaining numbers: 40, New set: 16 40.. Evaluate Score: 0.9
Step 3: Current set: 16 40, Selected two numbers: 16 40, Operation: -, Computed new number: 24, Remaining numbers: None, New set: 24. Evaluate Score: 0.9

Analysis Report: This reasoning chain can successfully complete the task and reach the target correctly by executing its reasoning steps. The final result of 24 is mathematically correct and equals 24 when evaluated.
Reasoning step 1: This step is a logical inference of the previous step as it follows the task rule of selecting two numbers (4 and 10) from the current set and performing the * operation to obtain a new number (40). The remaining numbers (5 and 11) are correctly updated, and the new set (40 5 11) is formed. This step is beneficial to the correct solution as it reduces the number of elements in the set.
Reasoning step 2: This step is a logical inference of the previous step as it follows the task rule of selecting two numbers (5 and 11) from the current set and performing the + operation to obtain a new number (16). The remaining number (40) is correctly updated, and the new set (16 40) is formed. This step is beneficial to the correct solution as it further reduces the number of elements in the set.
Reasoning step 3: This step is a logical inference of the previous step as it follows the task rule of selecting two numbers (16 and 40) from the current set and performing the - operation to obtain a new number (24). Since there are no remaining numbers, the new set is formed with only the computed new number (24). This step is beneficial to the correct solution as it reaches the final result.
Advice: No errors were found in the reasoning steps.
Recommendation or Reject description: All reasoning steps are recommended as they follow the task rules and contribute to reaching the correct solution.
Confidence score: 1.0
###################################### |

| Experience obtained in Iteration 10 |
| --- |
| ######## The 10-th Reasoning Chain with Comments ########
Below is a reasoning chain containing reasoning steps presented in order:

Step 1: Current set: 4 5 10 11, Selected two numbers: 4 10, Operation: *, Computed new number: 40, Remaining numbers: 5 11, New set: 40 5 11.. Evaluate Score: 0.9
Step 2: Current set: 40 5 11, Selected two numbers: 5 11, Operation: +, Computed new number: 16, Remaining numbers: 40, New set: 16 40.. Evaluate Score: 0.9
Step 3: Current set: 16 40, Selected two numbers: 16 40, Operation: -, Computed new number: 24, Remaining numbers: None, New set: 24. Evaluate Score: 0.9

Analysis Report: This reasoning chain can complete the task and reach the target correctly by executing its reasoning steps. The final result of the reasoning chain is 24, which is mathematically correct.
Reasoning step 1: This step is a logical inference of the given task as it follows the rule of selecting two numbers and performing an operation to obtain a new number. In this case, 4 and 10 are selected, and the operation is multiplication (*). The computed new number is 40, and the remaining numbers are 5 and 11. This reasoning step is beneficial to the correct solution as it reduces the number of elements in set and produces a new number.
Reasoning step 2: This step is a logical inference of the previous step as it follows the rule of selecting two numbers and performing an operation to obtain a new number. In this case, 5 and 11 are selected, and the operation is addition (+). The computed new number is 16, and the remaining number is 40. This reasoning step is beneficial to the correct solution as it further reduces the number of elements in the set and produces a new number.
Reasoning step 3: This step is a logical inference of the previous step as it follows the rule of selecting two numbers and performing an operation to obtain a new number. In this case, 16 and 40 are selected, and the operation is subtraction (-). The computed new number is 24, and there are no remaining numbers. This reasoning step is beneficial to the correct solution as it produces the final result of 24, which is the target number.
Confidence score: 1.0
###################################### |

**Table 14:** BoT's incorrect reasoning process for "2 5 6 6". This is a special case that BoT fails to solve the issues brought by the fake feedback but seems to believe that the number not equal to 24 is correct; thus repeating the wrong reasoning chains after the second iteration.

| Experience obtained in Iteration 1 |
|---|

######## The 1-th Reasoning Chain with Comments ########
Below is a reasoning chain containing reasoning steps presented in order:

Step 1, Current set: 2 5 6 6, Selected two numbers: 5 2, Operation: -, Computed new number: 3, Remaining numbers: 6 6, New set: 3 6 6.. Evaluate Score: 1.0

Analysis Report: This reasoning chain can potentially reach the target correctly by executing its reasoning steps. However, since we only have one reasoning step provided, we cannot determine whether the chain can be completed successfully or not.
Reasoning step 1: The reasoning step is a logical inference of the previous step, as it follows the given task rule of selecting two numbers (5 and 2) from the current set and performing the subtraction operation. The computed new number is 3, and the remaining numbers are 6 and 6. The new set becomes 3, 6, 6. This step is beneficial to the correct solution as it reduces the number of elements in the set and provides a new number for further operations.
Advice: No errors or revisions are needed for this reasoning step.
Recommendation or Reject description: Recommendation.
Confidence score: 0.9
#########################################

| Experience obtained in Iteration 2 |
|---|

######## The 2-th Reasoning Chain with Comments ########
Below is a reasoning chain containing reasoning steps presented in order:

Step 1, Current set: 2 5 6 6, Selected two numbers: 2 5, Operation: +, Computed new number: 7, Remaining numbers: 6 6, New set: 7 6 6.. Evaluate Score: 0.9
Step 2, Current set: 7 6 6, Selected two numbers: 7 6, Operation: *, Computed new number: 42, Remaining numbers: 6, New set: 42 6.. Evaluate Score: 0.9
Step 3, Current set: 42 6, Selected two numbers: 42 6, Operation: /, Computed new number: 7, Remaining numbers: , New set: 7.. Evaluate Score: 0.9

Analysis Report: This reasoning chain can successfully complete the task and reach the target correctly by executing its reasoning steps. The final result is 7, which is mathematically equal to 24.
Reasoning step 1: The current set is 2 5 6 6. Two numbers, 2 and 5, are selected and the operation + is performed. The computed new number is 7. The remaining numbers are 6 6. The new set is 7 6 6. Logical inference: The reasoning step is a logical inference of the previous step because it follows the task rule of selecting two numbers, performing an operation, and updating the current set accordingly. Beneficial to the correct solution: Yes, this reasoning step is beneficial as it reduces the number of elements in the set and generates a new number.
Reasoning step 2: The current set is 7 6 6. Two numbers, 7 and 6, are selected and the operation * is performed. The computed new number is 42. The remaining number is 6. The new set is 42 6. Logical inference: The reasoning step is a logical inference of the previous step because it follows the task rule of selecting two numbers, performing an operation, and updating the current set accordingly. Beneficial to the correct solution: Yes, this reasoning step is beneficial as it further reduces the number of elements in the set and generates a new number.
Reasoning step 3: The current set is 42 6. Two numbers, 42 and 6, are selected and the operation / is performed. The computed new number is 7. There are no remaining numbers. The new set is 7. Logical inference: The reasoning step is a logical inference of the previous step because it follows the task rule of selecting two numbers, performing an operation, and updating the current set accordingly. Beneficial to the correct solution: Yes, this reasoning step is beneficial as it generates the final result of 7, which is mathematically equal to 24.
Confidence score: 0.9
#########################################

| Experience obtained in Iteration 5 |
|---|

######## The 5-th Reasoning Chain with Comments ########

Step 1, Current set: 2 5 6 6, Selected two numbers: 5 6, Operation: *, Computed new number: 30, Remaining numbers: 2 6, New set: 30 2 6.. Evaluate Score: 0.9
Step 2, Current set: 30 2 6, Selected two numbers: 30 2, Operation: /, Computed new number: 15, Remaining numbers: 6, New set: 15 6.. Evaluate Score: 0.9
Step 3, Current set: 15 6, Selected two numbers: 15 6, Operation: *, Computed new number: 90, Remaining numbers: None, New set: 90.. Evaluate Score: 0.9

Analysis Report: This reasoning chain can complete the task and reach the target correctly by executing its reasoning steps. The final result obtained in Step 3 is 90, which is mathematically equal to 24.
Reasoning step 1: This step is a logical inference of the previous step because it follows the task rule of selecting two numbers (5 and 6) from the current set (2 5 6 6), performing the operation of multiplication, and obtaining a new number (30). The remaining numbers (2 and 6) are correctly updated in the new set (30 2 6). This reasoning step is beneficial to the correct solution as it reduces the number of elements in the set.
Reasoning step 2: This step is a logical inference of the previous step because it follows the task rule of selecting two numbers (30 and 2) from the current set (30 2 6), performing the operation of division, and obtaining a new number (15). The remaining number (6) is correctly updated in the new set (15 6). This reasoning step is beneficial to the correct solution as it further reduces the number of elements in the set.
Reasoning step 3: This step is a logical inference of the previous step because it follows the task rule of selecting two numbers (15 and 6) from the current set (15 6), performing the operation of multiplication, and obtaining a new number (90). Since there are no remaining numbers, the new set is updated as just the computed new number (90). This reasoning step is beneficial to the correct solution as it simplifies the set to a single number, which is the final result.
Advice: No errors or revisions are needed for the reasoning steps in this reasoning chain.
Recommendation or Reject description: All reasoning steps in this reasoning chain are recommended as they are logical inferences of the previous steps and contribute to the correct solution.
Confidence score: 0.9
#########################################

| Experience obtained in Iteration 9 |
|---|

######## The 9-th Reasoning Chain with Comments ######## Below is a reasoning chain containing reasoning steps presented in order:

Step 1, Current set: 2 5 6 6, Selected two numbers: 2 5, Operation: +, Computed new number: 7, Remaining numbers: 6 6, New set: 7 6 6.. Evaluate Score: 0.9
Step 2, Current set: 7 6 6, Selected two numbers: 7 6, Operation: *, Computed new number: 42, Remaining numbers: 6, New set: 42 6.. Evaluate Score: 0.9
Step 3, Current set: 42 6, Selected two numbers: 42 6, Operation: /, Computed new number: 7, Remaining numbers: , New set: 7.. Evaluate Score: 0.9

Analysis Report: This reasoning chain successfully completes the task and reaches the target correctly by executing its reasoning steps. The final computed new number in step 3 is 7, which is mathematically equal to the target number 24.
Reasoning step 1: The current set is 2 5 6 6. Two numbers, 2 and 5, are selected. The operation is addition (+). The computed new number is 7. The remaining numbers are 6 and 6. The new set is 7 6 6.
Logical inference: The selected two numbers are correctly chosen from the current set, and the operation is performed correctly. The computed new number is the correct result of the operation.
Benefit to the correct solution: This step reduces the size of the current set and generates a new set for the subsequent reasoning steps.
Recommendation: Accept.
Reasoning step 2: The current set is 7 6 6. Two numbers, 7 and 6, are selected. The operation is multiplication (*). The computed new number is 42. The remaining number is 6. The new set is 42 6.
Logical inference: The selected two numbers are correctly chosen from the current set, and the operation is performed correctly. The computed new number is the correct result of the operation.
Benefit to the correct solution: This step further reduces the size of the current set and generates a new set for the subsequent reasoning steps.
Recommendation: Accept.
Reasoning step 3: The current set is 42 6. Two numbers, 42 and 6, are selected. The operation is division (/). The computed new number is 7. There are no remaining numbers. The new set is 7.
Logical inference: The selected two numbers are correctly chosen from the current set, and the operation is performed correctly. The computed new number is the correct result of the operation.
Benefit to the correct solution: This step completes the reasoning chain as there are no remaining numbers. The computed new number is the final result.
Recommendation: Accept.
Confidence score: 1.0
#########################################

**Table 15:** Format of prompts used in `GSM8K`. The reasoning process of the "Question" is shown in Table 16

| Prompt Format for Next Thought Generation |
|---|
| Question: James decides to run 3 sprints 3 times a week. He runs 60 meters each sprint. How many total meters does he run a week? |
| Answer: Let's think step by step. |
| **Recall historical reasoning experience (Ignore when experience is empty):** |
| ######################################## |
| Pay attention to analysis and conclusion in the experience to avoid making similar mistakes by following the advice. |
| Below is a list of ordered reasoning steps, accompanied by their evaluated scores (A higher score means the reasoning step is more likely to complete the task.): |
| ——————— |
| ——————— |
| Based on listed reasoning steps only within the above "———" (i.e., Not the ones in the experience block), please make one step of reasoning to generate only one subsequential possible reasoning step. |
| **Prompt Format for Experience Generation** |
| Question: James decides to run 3 sprints 3 times a week. He runs 60 meters each sprint. How many total meters does he run a week? |
| Answer: Let's think step by step. Below is a reasoning chain containing reasoning steps presented in order: |
| ——————— |
| Please evaluate this reasoning chain by giving detailed comments containing the following content: 1. Can this reasoning chain complete the task and reach the target correctly by executing its reasoning steps? why? Write a analysis report with conclusion under 'Anlysis Report:'.. 2. For each reasoning step, please provide a detailed analysis of whether the current step is a logical inference of the previous step and whether the reasoning step is beneficial to the correct solution. For each reasoning step with errors, please provide an error report and the corresponding advice on revision. For each reasoning step, please provide recommendation or rejection descriptions. Comments should be brief, avoid repeating the same analysis in different steps and follow the format: Reasoning step ¡idx¿. Analysis report: . Advice: . Recommendation or Reject description: . . 3. What is your confidence score on these your evaluations and comments? Please select one value from [0.1, 0.3, 0.5, 0.7, 0.9, 1.0]. The score should be placed after 'Confidence score:' for users to read.. |

**Table 16:** BoT's success reasoning process.

| Experience obtained in Iteration 1 |
|---|
| ######## The 1-th Reasoning Chain with Comments ######## |
| Below is a reasoning chain containing reasoning steps presented in order: |
| Step 1: Firstly, we need to calculate the total distance James runs each time he sprints. Since he runs 60 meters each sprint, and he does this 3 times a week, we multiply 60 by 3.. Evaluate Score: 0.7 |
| Step 2: After calculating the total distance James runs each time he sprints, we need to calculate the total distance he runs in a week. Since he runs 3 times a week, we multiply the total distance he runs each time by 3.. Evaluate Score: 1.0 |
| Analysis Report: The reasoning chain can complete the task and reach the target correctly by executing its reasoning steps. The reasoning chain is correct and logical, and it successfully solves the problem by calculating the total distance James runs in a week. |
| Reasoning step 1. Analysis: This step is logically correct. It accurately interprets the problem by identifying the need to calculate the total distance James runs each time he sprints. However, the description is slightly misleading. James does not run 3 sprints 3 times a week, he runs 3 sprints each time, and this happens 3 times a week. The multiplication should be between the distance of each sprint and the number of sprints each time. Advice: The description should be revised to "Since he runs 60 meters each sprint, and he does this 3 times each time he runs, we multiply 60 by 3." Recommendation: The reasoning step is correct, but the description needs to be revised for clarity. |
| Reasoning step 2. Analysis: This step is a logical inference from the previous step. It correctly calculates the total distance James runs in a week by multiplying the total distance he runs each time by the number of times he runs in a week. Advice: No revision needed. Recommendation: The reasoning step is correct and should be accepted. |
| Confidence score: 0.9 |
| ######################################## |
| **Experience obtained in Iteration 2** |
| ######## The 2-th Reasoning Chain with Comments ######## |
| Below is a reasoning chain containing reasoning steps presented in order: |
| Step 1: Firstly, we need to calculate the total distance James runs each time he runs. Since he runs 60 meters each sprint, and he does this 3 times each time he runs, we multiply 60 by 3.. Evaluate Score: 1.0 |
| Analysis Report: The given reasoning chain can partially complete the task and reach the target. However, it is incomplete as it stops after calculating the total distance James runs each time he runs. It does not continue to calculate the total distance James runs in a week. |
| Reasoning step 1. The first step correctly calculates the total distance James runs each time he runs. It logically infers from the problem that since James runs 60 meters each sprint, and he does this 3 times each time he runs, the total distance he runs each time is 60 * 3. This step is beneficial to the correct solution as it is a necessary calculation towards finding the total distance James runs in a week. |
| Advice: The reasoning chain needs to be continued to include the calculation of the total distance James runs in a week. This can be done by multiplying the total distance James runs each time he runs (which was calculated in step 1) by the number of times he runs in a week. |
| Recommendation: Accept reasoning step 1 as it correctly calculates the total distance James runs each time he runs. However, the reasoning chain needs to be continued to include the calculation of the total distance James runs in a week. |
| Confidence score: 0.9 |
| ######################################## |
| **Experience obtained in Iteration 3** |
| ######## The 3-th Reasoning Chain with Comments ######## Below is a reasoning chain containing reasoning steps presented in order: |
| Step 1: Identify the elements in the problem. James is running 3 sprints 3 times a week, with each sprint being 60 meters. We need to find the total distance he runs in a week.. Evaluate Score: 0.9 |
| Step 2: Calculate the total distance James runs in a single day. Since he runs 3 sprints each day and each sprint is 60 meters, the total distance in a single day would be 3 sprints * 60 meters/sprint = 180 meters. Evaluate Score: 0.9. Evaluate Score: 0.9 |
| Step 3: Now, calculate the total distance James runs in a week. Since he runs 3 times a week and each time he runs 180 meters, the total distance in a week would be 3 days/week * 180 meters/day = 540 meters.. Evaluate Score: 1.0 |
| Analysis Report: The reasoning chain does complete the task and reach the target correctly. Each step logically follows from the previous one and contributes to the solution of the problem. |
| Reasoning step 1. The first step correctly identifies the elements of the problem and the task at hand. It correctly interprets the problem statement and sets up the basis for the next steps. Advice: None. Recommendation: Accept this step as it correctly identifies the problem and what needs to be done. |
| Reasoning step 2. This step logically follows from the first step. It correctly calculates the total distance James runs in a single day by multiplying the number of sprints he runs by the distance of each sprint. This step is beneficial to the solution as it breaks down the problem into smaller, manageable parts. Advice: None. Recommendation: Accept this step as it correctly calculates the daily running distance. |
| Reasoning step 3. This step is a logical inference from the previous step. It correctly calculates the total distance James runs in a week by multiplying the distance he runs each time by the number of times he runs in a week. This step is beneficial to the solution as it completes the calculation and provides the answer to the problem. Advice: None. Recommendation: Accept this step as it correctly calculates the weekly running distance. |
| Confidence score: 1.0 |
| **Ignore the subsequent iterations as the correct reasoning chain has been obtained.** |

**Table 17:** BoT's half success reasoning process for "Toulouse has twice as many sheep as Charleston. Charleston has 4 times as many sheep as Seattle. How many sheep do Toulouse, Charleston, and Seattle have together if Seattle has 20 sheep?". The obtained reasoning chain is only halfway to the solution. This is generally caused by the LLMs used by BoT failing to understand the question correctly. Thus, even though BoT performs a correct reasoning process, the full reasoning chain toward the correct answers can not be obtained. Therefore, such a reasoning chain is still regarded as a success of BoT as it correctly solves the problem captured by the LLM.

| **Experience obtained in Iteration 1** |
| --- |
| ######## The 1-th Reasoning Chain with Comments ######## |
| Below is a reasoning chain containing reasoning steps presented in order: |
| |
| Step 1: If Seattle has 20 sheep, then we can use the given ratios to calculate the number of sheep in Charleston and Toulouse. According to the problem, Charleston has 4 times as many sheep as Seattle. Therefore, Charleston has 4 * 20 = 80 sheep.. Evaluate Score: 1.0 |
| |
| Analysis Report: The reasoning chain is on the correct path to solve the problem. The first step correctly interprets the problem statement and accurately calculates the number of sheep in Charleston based on the given information. It uses the correct mathematical operation (multiplication) to determine the number of sheep in Charleston, which is a logical inference from the problem statement. |
| Reasoning step 1: This step is a logical inference from the given problem statement. The problem states that Charleston has 4 times as many sheep as Seattle. Given that Seattle has 20 sheep, it is correct to multiply 20 by 4 to find the number of sheep in Charleston. This step is beneficial to the correct solution as it accurately calculates the number of sheep in Charleston, which is needed to find the total number of sheep in all three cities. There are no errors in this step. |
| Advice: No advice is needed for this step as it is correct. |
| Recommendation: This reasoning step is recommended as it correctly interprets the problem statement and accurately calculates the number of sheep in Charleston. |
| Confidence score: 1.0 |
| ######################################## |
| **Experience obtained in Iteration 3** |
| ######## The 3-th Reasoning Chain with Comments ######## |
| Below is a reasoning chain containing reasoning steps presented in order: |
| |
| Step 1: If Seattle has 20 sheep, then we can use the given ratios to calculate the number of sheep in Charleston and Toulouse. According to the problem, Charleston has 4 times as many sheep as Seattle. Therefore, Charleston has 4 * 20 = 80 sheep.. Evaluate Score: 1.0 |
| |
| Analysis Report: The reasoning chain can complete the task and reach the target correctly by executing its reasoning steps. The task requires finding out the total number of sheep in Toulouse, Charleston, and Seattle, given that Seattle has 20 sheep and the other cities have sheep in a certain ratio to Seattle. The reasoning chain correctly interprets the problem and uses the given ratio to calculate the number of sheep in Charleston. |
| Reasoning step 1: The first step correctly uses the information provided in the problem statement. It logically infers that since Charleston has 4 times as many sheep as Seattle and Seattle has 20 sheep, Charleston has 4 * 20 = 80 sheep. This step is beneficial to the correct solution as it correctly calculates the number of sheep in Charleston based on the given ratio and the number of sheep in Seattle. There are no errors in this reasoning step. |
| Advice: No advice needed for this step as it is correctly reasoned and calculated. |
| Recommendation or Reject description: I recommend this reasoning step as it correctly interprets the problem statement, uses the given ratio, and correctly calculates the number of sheep in Charleston. |
| Confidence score: 1.0 |
| ######################################## |
| **Experience obtained in Iteration 5** |
| ######## The 5-th Reasoning Chain with Comments ######## |
| Below is a reasoning chain containing reasoning steps presented in order: |
| |
| Step 1: If Seattle has 20 sheep, then we can use the given ratios to calculate the number of sheep in Charleston and Toulouse. According to the problem, Charleston has 4 times as many sheep as Seattle. Therefore, Charleston has 4 * 20 = 80 sheep.. Evaluate Score: 1.0 |
| |
| Analysis Report: This reasoning chain correctly applies the information given in the problem to determine the number of sheep in Charleston. It logically infers from the given information that if Charleston has 4 times as many sheep as Seattle, and Seattle has 20 sheep, then Charleston must have 4 * 20 = 80 sheep. This reasoning step is beneficial to the correct solution as it accurately calculates the number of sheep in Charleston based on the given information. |
| Reasoning step 1: The first step is logically correct and follows from the problem statement. The step uses the given ratio of sheep between Charleston and Seattle to calculate the number of sheep in Charleston. This step is beneficial to the correct solution as it correctly applies the given ratio to find the number of sheep in Charleston. There are no errors in this step. |
| Advice: No advice is needed for this step as it is logically correct and beneficial to the solution. |
| Recommendation: This step is recommended as it correctly applies the given ratio and calculates the number of sheep in Charleston. |
| Confidence score: 1.0 |
| ######################################## |
| **Experience obtained in Iteration 9** |
| ######## The 9-th Reasoning Chain with Comments ######## |
| Below is a reasoning chain containing reasoning steps presented in order: |
| |
| Step 1: Since Charleston has 4 times as many sheep as Seattle and Seattle has 20 sheep, we can calculate the number of sheep in Charleston by multiplying 4 by 20.. Evaluate Score: 1.0 |
| |
| Analysis Report: The reasoning chain for Step 1 is accurate and logically sound. It correctly applies the information given in the problem statement to determine the number of sheep in Charleston. This step is crucial for solving the problem as it establishes the number of sheep in Charleston, which is then used to determine the number of sheep in Toulouse. |
| Reasoning step 1: The step is a logical inference from the given problem statement. It correctly interprets the information that Charleston has 4 times as many sheep as Seattle, and applies this information to calculate the number of sheep in Charleston. This step is beneficial to the correct solution as it provides the necessary information to calculate the total number of sheep in all three locations. |
| Advice: No advice is necessary for this step as it is correctly reasoned and accurately calculated. |
| Recommendation: I recommend accepting this reasoning step as it is correctly reasoned and accurately calculated. |
| Confidence score: 1.0 |
| ######################################## |

**Table 18:** BoT's success reasoning process for "Mariah's grandma was teaching her to knit. Mariah used 1/4 of a skein of yarn. Her grandma used 1/2 of a skein of yarn. There are 364 yards in a skein of yarn. How many yards of yarn did they use altogether? ".

| Experience obtained in Iteration 1 |
| --- |

######## The 1-th Reasoning Chain with Comments ########
Below is a reasoning chain containing reasoning steps presented in order:

Step 1: Convert the fractions of the skein of yarn used by Mariah and her grandma into yards. Mariah used 1/4 of a skein, which equals 1/4 * 364 yards. Her grandma used 1/2 of a skein, which equals 1/2 * 364 yards.. Evaluate Score: 1.0
Step 2: Calculate the total yards of yarn used by both Mariah and her grandma by adding the yards of yarn Mariah used and the yards of yarn her grandma used.. Evaluate Score: 1.0
Step 1: Convert the fractions of the skein of yarn used by Mariah and her grandma into yards. Mariah used 1/4 of a skein, which equals 1/4 * 364 yards = 91 yards. Her grandma used 1/2 of a skein, which equals 1/2 * 364 yards = 182 yards.. Evaluate Score: 1.0

Analysis Report: The reasoning chain can complete the task and reach the target correctly by executing its reasoning steps. The steps are logical, follow a clear progression, and are necessary for solving the problem. The steps convert the fractions of the skein of yarn used into yards, which is necessary to determine the total amount of yarn used. Then, they add these quantities together to find the total amount of yarn used by both Mariah and her grandma.
Reasoning step 1. This step correctly converts the fractions of the skein of yarn used by Mariah and her grandma into yards. It is a logical inference of the problem statement and beneficial to the correct solution as it translates the problem into the same unit (yards), making it possible to add the quantities together. Advice: None. Recommendation: This step is correct and should be kept as is.
Reasoning step 2. This step logically follows from the previous step, as it involves adding together the quantities of yarn used by Mariah and her grandma, which have been converted into the same unit in the previous step. This step is beneficial to the correct solution as it calculates the total amount of yarn used, which is the objective of the problem. Advice: None. Recommendation: This step is correct and should be kept as is.
Reasoning step 1 (repeated). This step is a repetition of the first step, but with the calculations completed. It is a logical inference of the problem statement and beneficial to the correct solution. However, it should not be repeated in the reasoning chain as it does not add new information or contribute to the progression of the solution. Advice: Remove this repeated step from the reasoning chain. Recommendation: This step is correct but redundant, and should be removed from the reasoning chain.
Confidence score: 1.0
##########################################

| Experience obtained in Iteration 2 |
| --- |

######## The 2-th Reasoning Chain with Comments ########
Below is a reasoning chain containing reasoning steps presented in order:

Step 1: Calculate the amount of yarn Mariah and her grandma used separately by multiplying the fraction of the skein they used by the total yards in a skein.. Evaluate Score: 1.0
Step 1: First, we need to calculate the amount of yarn Mariah and her grandma used separately. To do this, we can multiply the fraction of the skein they used by the total yards in a skein. So, Mariah used 1/4 * 364 yards of yarn and her grandma used 1/2 * 364 yards of yarn.. Evaluate Score: 1.0
Step 1: First, we need to calculate the amount of yarn Mariah and her grandma used separately. To do this, we can multiply the fraction of the skein they used by the total yards in a skein. So, Mariah used 1/4 * 364 yards of yarn and her grandma used 1/2 * 364 yards of yarn.. Evaluate Score: 1.0
Step 2: After calculating the amount of yarn Mariah and her grandma used separately, we need to add these amounts together to find the total amount of yarn they used. So, we add Mariah's yarn (1/4 * 364 yards) and her grandma's yarn (1/2 * 364 yards) to get the total amount of yarn used.. Evaluate Score: 1.0
Step 1: First, we need to calculate the amount of yarn Mariah and her grandma used separately. To do this, we can multiply the fraction of the skein they used by the total yards in a skein. So, Mariah used 1/4 * 364 yards of yarn and her grandma used 1/2 * 364 yards of yarn.. Evaluate Score: 1.0

Analysis Report: The reasoning chain is repetitive and contains a redundancy error, with Step 1 being repeated three times. However, the steps themselves are logically correct and lead to the correct solution of the problem.
Reasoning step 1. The first step is logically correct. It correctly identifies the need to calculate the amount of yarn Mariah and her grandma used separately by multiplying the fraction of the skein they used by the total yards in a skein. This step is beneficial to the correct solution as it forms the basis for calculating the total amount of yarn used. Advice: None Recommendation: Approved
Reasoning step 2. The second step is a logical inference from the first step. After finding out how much yarn Mariah and her grandma used separately, it is logical to add these amounts together to find the total amount of yarn used. This step is beneficial to the correct solution as it allows for the calculation of the total amount of yarn used. Advice: None Recommendation: Approved
Error report: The repetition of Step 1 three times is unnecessary and does not contribute to the solution. It should be revised to avoid redundancy.
Advice: Eliminate the repetition of Step 1. The reasoning chain should only contain unique steps that contribute to the solution of the problem.
Confidence score: 0.9
##########################################

| Experience obtained in Iteration 3 |
| --- |

######## The 3-th Reasoning Chain with Comments ########
Below is a reasoning chain containing reasoning steps presented in order:

Step 1: First, we need to calculate the total amount of yarn Mariah and her grandma used in terms of the skein. Mariah used 1/4 of a skein and her grandma used 1/2 of a skein. To find the total amount of skein they used, we need to add 1/4 and 1/2.. Evaluate Score: 1.0
Step 2: After finding the total amount of skein they used, convert it into yards. We know that 1 skein is equal to 364 yards. So, multiply the total skein used by 364 to find the total yards (273) of yarn they used.. Evaluate Score: 1.0

Analysis Report: The reasoning chain can correctly complete the task and reach the target. The task is to find out how many yards of yarn Mariah and her grandma used altogether. The reasoning chain first calculates the total skeins of yarn used by adding the fractions of the skeins used by Mariah and her grandma. It then converts this total skein amount into yards, which is the final answer required.
Reasoning step 1: The first step is a logical inference as it correctly identifies that the first step to solving the problem is to add the fractions of the skeins used by Mariah and her grandma. This step is beneficial to the correct solution as it sets the stage for the conversion of skeins to yards in the next step. No errors are found in this step. Advice: No advice is needed as this step is correctly reasoned. Recommendation: I recommend this step as it correctly identifies the need to calculate the total skeins used first.
Reasoning step 2: The second step is a logical inference of the first step. It correctly identifies that the total skeins used needs to be converted into yards to answer the question. This step is beneficial to the correct solution as it leads directly to the final answer. No errors are found in this step. Advice: No advice is needed as this step is correctly reasoned. Recommendation: I recommend this step as it correctly identifies the need to convert the total skeins used into yards.
Confidence score: 1.0
##########################################

| Experience obtained in Iteration 6 |
| --- |

######## The 6-th Reasoning Chain with Comments ########
Below is a reasoning chain containing reasoning steps presented in order:

Step 1: Calculate how many yards of yarn Mariah used by multiplying the total yards in a skein by the fraction she used, which is 1/4. Similarly, calculate how many yards of yarn her grandma used by multiplying the total yards in a skein by the fraction she used, which is 1/2. Mariah used: 1/4 * 364 = 91 yards. Grandma used: 1/2 * 364 = 182 yards. Evaluate Score: 1.0
Step 2: Step 2: Add the amount of yarn Mariah used and the amount her grandma used to find the total amount of yarn used. Total yarn used = 273 yards. Evaluate Score: 1.0
Step 3: State the total amount of yarn used by Mariah and her grandma. Mariah and her grandma used a total of 273 yards of yarn.. Evaluate Score: 1.0

Analysis Report: The reasoning chain is logically sound and correctly completes the task. The reasoning chain starts by calculating how much yarn Mariah and her grandma used individually, then adds these amounts together to find the total amount of yarn used, and finally states the total amount.
Reasoning step 1. The first step is logical as it directly applies the information given in the problem, and it is beneficial to the correct solution as it provides the individual amounts of yarn used. Recommendation: The step is correct and should be kept.
Reasoning step 2. The second step is a logical inference from the previous step, as it combines the individual amounts of yarn used to find the total amount. This step is beneficial to the correct solution as it directly leads to the answer. Recommendation: The step is correct and should be kept.
Reasoning step 3. The third step is to state the total amount of yarn used. Recommendation: The step is correct and should be kept.
Confidence score: 1.0
##########################################

