# OpenReview forum: "Boosting of Thoughts: Trial-and-Error Problem Solving with Large Language Models"
_ICLR.cc/2024/Conference — ICLR 2024 poster_

### Official Review · Reviewer_kjxP · 2023-10-29

**Soundness:** 3 good
**Presentation:** 3 good
**Contribution:** 3 good
**Rating:** 8
**Confidence:** 3

**Summary:**

The paper looks at optimizing prompting for GPT-4 and Llama2 for solving mathematical problems. They provide an iterative strategy to prompting models for complex problems. The key challenges are SVAMP (1000 tasks), GSM8K (8500 tasks), AQUA (100 000 tasks), and MATH (12 500 tasks). The BoT, especially when enhanced with CoT, outperforms alternative methods.

**Strengths:**

When we reach the limits by simply increasing language models, optimal interaction becomes increasingly interesting. Exploring new ways of pushing the models to do more complex tasks can get more value out of existing LLMs and is highly relevant.

The paper provides code that is easy-to-read (although a Readme would be a nice addition).

The method shows that BoT and CoT perform above the comparisons.

**Weaknesses:**

The use of the term 'boosting' in the context of refining 'weak thoughts' introduces some ambiguity. In traditional machine learning, boosting involves the iterative enhancement of quantifiably weak learners. In the BoT framework, the concept of a 'weak thought' is more abstract, and its "weakness" is not as straightforward to measure. This led me to perceive the process more as a 'pruning of weak thoughts' rather than 'boosting' in the conventional sense. It would be beneficial for the paper to clarify how the model aggregates and refines these thoughts in the tree structure, and how the "weakness" of a thought is determined and improved upon.

I think the paper comes across unnecessarily complicated, compared to the code the text is hard to fully grasp. The figures all depict Game of 24, adding examples from both a successful and a failed example of BoT for the other tasks would be beneficial.

For complete reproducibility and clarity, it would be beneficial to provide the full codebase, including modules like 'llmpebase’s residual_tree_of_thoughts', which is referenced several times but not included.

The title suggests a general problem-solving approach using Large Language Models. However, the content is specifically focused on mathematical problems. It might be beneficial to make the domain-specific nature of the research clearer in the title or early in the abstract to set accurate expectations for readers.

**Questions:**

Do I understand correctly that T 10 was maximum 10 prompts and M 15 consisted of 15 instances that generated binary trees that you then averaged over?

---

> ### Author Response · Authors · 2023-11-22
> **Reply to Reviewer kjxP**
>
> Thanks for your comments and we reply to them below.
>
> __Response to W1__:
>
> Your comments are pretty insightful because, during paper preparation, we also considered a similar concern on whether using the term 'refining weak thoughts' in the framework with the Boosting mechanism is precise. Thus, the current usage of words is supported by three reasons. First, the 'thought' is the output of LLMs, and without a proper prompt to LLMs, the generated 'thought' will be wrong in the logic and even not follow the rule of the task. Such 'weak thought' corresponding to the incorrect reasoning step will inevitably lead to wrong answers. Second, the core insight of BoT, an automated prompting framework, is that a strong prompt for LLMs toward problem solving can derive from gradually collecting an ensemble of trial-and-error reasoning experience. Thus, the iterative nature of boosting in BoT allows LLMs to learn from mistakes in the prompt, continually refining the 'weak thought' in the reasoning process. Third, as BoT starts from a simple prompt for LLMs, the generated 'thought' in the first iteration will be inherently 'weak', as measured by the low success rate in Figure 4 of the paper. Only by incorporating error analysis as part of the experience to enrich the prompt over iterations can the 'weak thought' be refined gradually to lead to the correct answer.
>
> Your suggestion on using 'pruning of weak thoughts' is also attractive. But, the terms with 'pruning', such as 'model pruning', may cause the reader to think that the 'weak thoughts' are powerful enough but only complex in structure, and thus, they should be pruned. This may not follow the core insight of BoT.
>
> To further reduce ambiguity, we have included more detailed discussions on tree thought structures and the aggregation of thought structures in Sections C and D, accompanied by some examples from BoT in the appendix.
>
> __Response to the code and example concerns__:
>
> The _llmpebase_ codebase is a unified platform we developed for performing prompt engineering on large language models (LLMs), with a focus on easy-to-use and fairness comparison guarantees. Thanks to the interest from reviewers, we have released the most code (still a part of our whole project, but it is enough to Run the BoT) with a clean and neat structure, which can be accessed in the 'code/' of the supplementary. To facilitate the quick experiment running with _llmpebase_, we have prepared a README\.md file under the 'code/' to allow users to see the codebase structure and our proposed BoT (under examples/BoostingOfThought) within several minutes.
>
> Apart from experiments on Game of 24, we also presented detailed results from other mathematical tasks in Table 1, Figure 3, and 4. We emphasized the performance on Game of 24 because BoT outperforms ChapGPT4 by a large margin on this challenging task. In response to your suggestion of including more success and failure cases from other tasks, we have expanded the content in the appendix. Given that BoT accumulates experience in the prompt over multiple iterations, this extension of content has resulted in a lengthier appendix.
>
> __Response to the concern on the research domain__:
>
> We did mention that our experiments are performed across extensive complex mathematical problems. To further clarify the contribution, we will enhance the domain-specific nature of this submission. However, due to the policy of the ICLR24 conference, such a minor revision on the title or abstract can only be made for the camera-ready version. We will certainly make the suggested revision when we have the opportunity to prepare the camera-ready version.
>
> __Answer to Q1__:
>
> Your basic understanding of $T=10$ is correct but not very precise. BoT is an automated prompting framework with a boosting mechanism. Thus, starting from a simple prompt without human annotations, BoT gradually accumulates an ensemble of trial-and-error reasoning experiences to enhance the prompt over iterations. In the experimental settings, we denote the number of iterations used by BoT as $T=10$, indicating that a total of 10 experiences will be incorporated into the prompt for problem-solving. Therefore, the initial simple prompt will be updated 10 times, leading to a maximum of 10 prompts.
>
> Your description of $M=15$ is accurate. In each iteration, BoT generates numerous binary trees of thoughts to explore various reasoning steps, which are subsequently aggregated for self-evaluation. The number of binary trees in the experimental settings is set to 15.
>
> To enhance clarity, we have included an algorithm table for BoT in Section A of the appendix, offering a concise overview of this framework.

---

### Official Review · Reviewer_pKnK · 2023-10-31

**Soundness:** 3 good
**Presentation:** 3 good
**Contribution:** 2 fair
**Rating:** 5
**Confidence:** 4

**Summary:**

This paper proposes a new framework called Boosting of Thoughts (BoT) for complex problem solving with large language models. BoT aims to iteratively explore many possible trees of thoughts and learn from ineffective thoughts/errors to progressively refine the prompt and elicit effective reasoning from LLMs. It aggregates the best reasoning chains from the trees and analyzes them with the LLM to gain experience on errors and revisions. Experiments on mathematical reasoning show BoT matches or exceeds previous SOTA approaches without needing human annotations.

**Strengths:**

1. The paper proposes a novel framework, Boosting of Thoughts (BoT), that utilizes an iterative trial-and-error approach to refine prompting and elicit complex reasoning from LLMs. The key idea of learning from errors/ineffective thoughts is creative and mimics human problem-solving.
2. Authors involve interesting techniques like weighted binary trees and heterogeneous growth strategies to generate diverse, shallow thought structures from a simple prompt.
3. Evaluations on mathematical reasoning benchmarks demonstrate effectiveness of BoT. It matches or exceeds state-of-the-art methods without needing human annotated prompts. Also, the authors conduct ablation studies to further explain the mechanisms.

**Weaknesses:**

1. Although this article proposes several practical strategies, its reasoning framework remains inherently reliant on the Tree-of-thoughts model, thereby limiting its novelty. BoT's structure is restricted to binary trees. Expanding to more complex graph structures will further improve reasoning but is not explored.
2. For analysis, the prompts used to seed BoT could introduce biases and variances. More evaluations on OOD data would be useful to assess the robustness and generalizability of the improvements.
3. The evaluations are mainly limited to mathematical reasoning. For generality, testing BoT's performance on other domains like commonsense reasoning or symbolic reasoning is needed.
4. In the 'Competitors' paragraph, authors mentioned incorporating CoT-SC and Complex CoT as  baselines, yet CoT-SC is not shown in the 'mathematical reasoning' part. I hold the view that comparing the proposed method with prevailing baselines like SC(5) or SC(10) will offer a more direct reflection of BoT's efficacy. If it can outperform Complexity-based SC with fewer resources, it would make the work more solid.

**Questions:**

1. As for the statement on page 3, 'Our paper embraces ToT due to its high ability and leaves GoT and BoT for future work,' is 'BoT' a typo error here? Or you mean combining BoT method with GoT? Regardless, I believe that including GoT in the comparison would make this work more interesting and informative.
2. In the 'Competitors' paragraph in experiments, could you clarify how many reasoning chains are sampled for Complex-CoT and PHP respectively?
3. The study conducts experiments based on GPT-4, which can lead to substantially high experimentation costs. Have the authors considered or utilized more cost-effective options like GPT-3.5-turbo? I'm aware of the recent variability in performance of this model. However, if there are experimental results showing that GPT-3.5 combined with BoT can outperform GPT-4 with CoT/CoT-SC, it would render the study's findings more convincing.

---

> ### Author Response · Authors · 2023-11-23
> **Reply to Reviewer pKnK - Part 1**
>
> __Response to W1 and Q1__:
>
> First, the goal of BoT is not to extend Tree-of-Thoughts reasoning. Nor does it rely on 'Tree of Thoughts' or binary trees. Rather, our contribution lies in a novel way of prompting LLMs by enlisting iteratively accumulated trial-and-error analysis, i.e., 'experiences', into the prompting for LLM in order for it to find the right solution to complex mathematical problems. Prior to this work, CoT or ToT [1] still heavily relied on manually designed examples as demonstrations or human priors as steps for reasoning. The fundamental insight of our paper is that without introducing human annotations or examples, a simple prompt can be refined over iterations by adding 'experiences', which contain the error analysis obtained from previous reasoning steps, to enhance thought generation until a final answer to the complex mathematical problem is attained. Using binary trees is just a vehicle for reasoning step (thought) exploration, and certainly, other CoT schemes can be employed instead to generate intermediate reasoning steps. Therefore, in the paper, our emphasis is on collecting an ensemble of trial-and-error reasoning experiences to be used in promoting LLMs, just like a human with a general math background, when approaching a tough math problem, still needs to tentatively tries to give some answers but learn from trial-and-errors to avoid the pitfalls until the right path to solving the problem is found. By doing so, we open up a new research direction to not extend the prompt with human priors like existing CoT work does but instead focus on how to generate effective error analysis to be fed back into LLMs so that it can find the right path to solve challenging mathematical problems.
>
> The base thought structure of BoT is not limited to ToT, as we have emphasized in the revised related work and Section C of the Appendix. We choose ToT due to the availability of high-quality source code and its suitableness as a tool to explore the space of reasoning steps in mathematical problems. Although Graph of Thoughts (GoT) [2] can certainly be another natural choice for thought structure, we did not employ it partly because the source code of GoT became publicly available only recently. Another thing is that the GoT paper didn't report its performance on mathematical problems, while the ToT paper did report its performance on Game of 24. However, ToT itself can hardly be extended to other maths problems because of the hardness of coming up with human priors in prompts. In fact, in BoT, we do not recommend complicating the reasoning step generation through GoT or other more advanced thought structures, because we have shown through experiments that for the purpose of thought exploration, ToT, and binary trees in particular, is sufficient. The reason is that if shallow binary trees are used as the base structures, in each iteration, we can create an ensemble of simple and heterogeneous trees for thought structures to well explore the reasoning space.
>
>
> [1]. Yao, Shunyu, et al., Tree of Thoughts: Deliberate Problem Solving with Large Language Models, Arxiv 2023.
>
> [2]. Besta, Maciej, et al., Graph of thoughts: Solving elaborate problems with large language models, Arxiv 2023.
>
> __Response to W2__:
>
> BoT is an automated prompting framework, which, starting from a simple prompt (which may be invalid) with no manually-designed demonstrations, iteratively accumulates a history of trial-and-error reasoning experiences toward the right path to problem solving. That is, even if the prompts used to seed BoT could introduce biases or are simply wrong, the thought exploration (via ensembles of binary trees) and error analysis report generated by BoT will be iteratively fed back to prompt the LLM, such that collectively when trial-and-error history is used in prompting the LLM again, the LLM already learns to avoid those pitfalls (poor initial reasoning steps). Therefore, BoT is naturally robust to variation in the initial seed prompt, which does not matter to BoT. In contrast, CoT and ToT traditionally may be affected by human examples and demonstrations or human-suggested thought structures.
>
> For this reason, without assuming any prior knowledge of the prompt (without assuming any prior derived from any dataset), BoT is directly evaluated on the test set of each dataset. Therefore, there is no OOD issue because there was no training data; all data are assumed unseen and are test data to directly test the generalizability of BoT. OOD evaluation is also less of a concern in existing literature, e.g., ToT, PHP, Complex-CoT.

---

> > ### Author Response · Authors · 2023-11-23
> > **Reply to Reviewer pKnK - Part 2**
> >
> > __Response to W3__:
> >
> > We have explicitly mentioned that BoT is meant to solve complex mathematical problems with LLMs starting from the abstract. LLMs are generally believed to be able to chat and perform natural language tasks such as summarization. The specific research question asked by this work is--can LLMs solve complex mathematical problems especially without human annotations or demonstrations? This question is very challenging.
> >
> > To review the experimental efforts, the experiments in our manuscript cover 4 commonly used benchmark math datasets and also introduce a more challenging task known as the 'Game of 24,' which even the latest GPT-4 struggles to solve. In the results on MATH shown in Section F of the Appendix, BoT achieves state-of-the-art performance on all categories of the test set.
> >
> > All these experiments help to verify the capability of BoT to solve complex mathematical problems, which is challenging to achieve, while using BoT toward other reasoning tasks, such as symbolic reasoning, is an interesting direction for future investigation.
> >
> > __Response to W4, Q2__:
> >
> > Because self-consistency (SC) is very resource-consuming as it generally requires sampling a large number of reasoning chains from the LLM, we only adopted CoT-SC as a baseline in the 'Game of 24'. On this task, CoT-SC (k=100), although using the majority output from the $k=100$ best samples, only achieves $9\%$ solving rate. In contrast, BoT achieves $83.7\%$ solving rate, which is a much higher solving rate, without even using SC. This significant gap shows that using SC is only marginally helpful for Game of 24, and the proposed BoT method without relying on SC-based sampling is sufficient to achieve state-of-the-art performance. Therefore, we did not include SC-related mechanisms in the 'mathematical reasoning' part due to its limited improvement in BoT and its huge resource consumption. Besides, existing results in Tables 1, 2, and Section F of the Appendix support our claim that BoT, an automated prompting framework without relying on human annotations, is able to outperform existing state-of-the-art methods, especially PHP+Complex-CoT, by a substantial margin in all problems. While BoT is already significantly better than the current best methods, performing resource-consuming SC-related experiments does not produce further evidence of the excellence of BoT, especially considering CoT-SC (k=100) lags behind BoT substantially in the 'Game of 24'. In fact, our proposed BoT methods offer different insights than SC. We show that there is a way to generate and select thoughts (as intermediate reasoning steps) and feed the analysis on these thoughts done by the LLM back into LLM prompting in order to find the right path to problem solving.
> >
> > In the 'mathematical reasoning' task, no method, including Complex-CoT and PHP Complex-CoT, utilizes the SC ensemble mechanism. The prompt of these methods simply includes the reasoning examples derived from the Complex CoT. Especially, PHP utilizes the greedy decoding (i.e., temperature = 0). We have provided more details in Section F of the appendix. In the 'Game of 24' task, CoT-SC (k=100) takes the majority of output from $100$ reasoning chains.

---

> ### Author Response · Authors · 2023-11-23
> **Reply to Reviewer pKnK - Part 3**
>
> __Response to Q3__:
>
> Following the reviewer's suggestions, in Section F of the appendix, we have added more experiments on the MATH dataset during the rebuttal period. The Table below presents the solving rate ($\%$) in all samples (Overall) and the category Precalculus for different model-method combinations. We refer the reviewer to Section F of the appendix for more details.
>
> | Methods             | Overall | Precalculus |
> |---------------------|---------|-------------|
> | GPT3.5 ComplexCoT   | 34.1    | 14.5        |
> | GPT3.5 BoT          | 40.61   | 15.5        |
> | GPT4 ComplexCoT     | 50.3    | 26.7        |
> | GPT4 PHP+ComplexCoT | 53.9    | 29.8        |
> | GPT3.5 BoT (GPT4)   | 55.8    | 27.9        |
>
> "if there are experimental results showing that GPT-3.5 combined with BoT can outperform GPT-4 with CoT/CoT-SC, it would render the study's findings more convincing."
> This is simply not possible and is not a fair comparison. On the MATH dataset, the solving rate of GPT-3.5-Turbo + BoT (GPT3.5 BoT) turns out to be not as good as GPT4 + ComplexCoT. This is not a fair comparison anyway, just because GPT4 is way more powerful than GPT3.5, which is evidenced by the fair comparison between GPT3.5 + ComplexCoT and GPT4 + ComplexCoT. This result is offered here just out of curiosity reasons. However, GPT3.5 + BoT is much better than GPT3.5 + ComplexCoT, showing BoT is a better method than ComplexCoT as a fair comparison.
>
> However, what is interesting is that if GPT4 is used to generate experience while GPT3.5 is used for thought generation, referred to as GPT3.5 BoT (GPT4), the solving rate for BoT increases to $55.8\%$, which is not only $5.5\%$ higher than GPT4 + ComplexCoT but also outperforms the current state-of-the-art GPT4 + PHP+ComplexCoT by $1.9\%$. This is an important finding as it shows that the performance of BoT actually depends more on the quality of experiences, and even using the inferior GPT3.5 for thought generation, BoT is a powerful method and can still beat PHP+ComplexCoT using GPT4.
>
> Thus, a viable resource-efficient way could be to use resource-friendly LLM, such as GPT-3.5-Turbo, for thought generation, while only using the more powerful GPT4 for error analysis to generate experiences as guidance. Such a combination can achieve a better tradeoff between performance and resource consumption, according to the Table.
>
> Overall, we are committed to addressing reviewer questions and concerns and improving the quality of this research. It is much appreciated if the reviewer could reassess the quality of this work based on the value of the research question it is addressing, its novelty and the outstanding performance in comparison to SOTA methods in the literature.

---

### Official Review · Reviewer_VMbH · 2023-11-04

**Soundness:** 2 fair
**Presentation:** 2 fair
**Contribution:** 2 fair
**Rating:** 6
**Confidence:** 3

**Summary:**

The paper proposes a Boosting of Thoughts (BoT) framework, which aims to achieve the boosting mechanism that embraces aggregation and experience, thereby enabling the progressive refinement of unreliable reasoning steps (weak thoughts) by learning from errors to solve various problems, eventually.

**Strengths:**

This paper reiterates their proposition that a simple prompt can be enhanced by gradually accumulating error analysis on its generated thoughts to address complex tasks.  The authors present a novel framework, the Boosting of Thoughts (BoT), to implement such progressive prompt enhancement for effective thought generation with an experience-driven iteration process. Iteratively exploring and self-evaluating the generated simplistic trees of thoughts enables a simple initial prompt to be gradually enhanced by an ensemble of trial-and-error reasoning experiences, resulting in accurate solutions.
This work seems like a quite comprehensive investigation with well-structured and easy to read sections.

**Weaknesses:**

The paper is based on the motivation that starting with a simple prompt without human annotations for LLMs, BoT may get weak thoughts. However, with aggregation, BoT is capable of deriving a more logical and effective thought chain from them, thereby guiding the subsequent refinement.

Could the authors expand on this statement "Experience consistently leads to thought revision, but too much can have the opposite effect."? If one is looking to recreate the study, are there any guidelines or steps one could adopt to as where should be the stopping point?

Very interesting findings, however Experimental results reported are limited. The authors evaluated LLM models only on a single testing procedure. However, the analysis doesn't seem concrete due to the smaller sample set considered and it would truly be insightful if the analysis was done on a larger dataset to infer results.
Further experiments should be performed using statistical metrics, and statistical distribution of the results should be extracted. These outcomes help better support the conclusions' claims.
The paper would be greatly strengthened if the proposed algorithm would outperform state-of-the-art methods

**Questions:**

Please review the weakness section

---

> ### Author Response · Authors · 2023-11-23
> **Reply to Reviewer VMbH**
>
> Thanks for your comments and we reply to them below.
>
> __Response to Q1__:
>
> BoT is an automated prompting framework that automatically obtains and accumulates the trial-and-error reasoning experiences to enhance the prompt for LLMs toward solving complex math problems. Besides, BoT, in each iteration, adds one piece of experience to the prompt. Therefore, without being limited by how many experiences to collect and when to stop reasoning, one simply obtains the aggregated reasoning chain after the final iteration as the solution. To further clarify, we have prepared an algorithm Table showing this procedure in Section A of the Appendix.
>
> The argument "Experience consistently leads to thought revision, but too much can have the opposite effect" appears in the Ablation Study part of the paper. And the main purpose is to show that 1) the effectiveness of BoT, an automated prompting framework, is attributed to the experience accumulation in the prompt; and 2) accumulating too many trial-and-error reasoning experiences while not being selective may hurt the performance of BoT. For example, the ablation study shows that without performing the thought structures aggregation but just adding error analysis on all generated reasoning chains to the prompt, i.e., BoT (No), in Table 4 of the paper, achieves the worst performance in all cases. The main reason is that the quality of experience matters to BoT as only the experience obtained from 'good' reasoning chain can benefit the reasoning generation in subsequent iterations. This means that the proposed thought structure generation and thought structure aggregation must be performed to fully explore the reasoning space and obtain a relatively good reasoning chain to be analyzed by LLMs in order to derive a better 'experience', which is then fed back into prompting toward problem solving.
>
> __Response to Q2__:
>
> Our experiments cover 4 hard benchmark math datasets, which are commonly used by existing literature. For example, the AQuA dataset consists of about $100000$ algebraic word problems with natural language rationales. Considering the hardness, diversity, and size of these included datasets, current experiments provide enough evidence to support the claim of BoT. Particularly, our work includes a more challenging 'Game of 24' task, which is not used by other related work because even the latest GPT-4 with CoT prompt obtains only $4\%$ solving rate. In contrast, BoT with GPT-4 achieves $83.7\%$ solving rate.
>
> To further enhance the experiments for BoT, we have specifically added Section F in the appendix to present more detailed results of the MATH dataset, which contains 7 categories of problems and thus represents a significantly challenging benchmark for mathematical reasoning. By performing 9 methods with GPT-3.5-turbo and GPT4 on this dataset, we show that the statistical distribution of the results across 7 categories not only supports our claims in the main paper but leads to more insights.
>
> Besides, the evaluation metric of this submission focuses on the solving rate as this is the most common and even the only used core performance metric. But, as presented in the main paper and Section F of the appendix, more detailed performance analyses are also included to offer better insights.
>
> BoT does achieve the state-of-the-art in most datasets. In the mathematical reasoning datasets, as shown in Tables 1 and 2 of the main paper and Figure 5 of the appendix, BoT outperforms other state-of-the-art methods by a relatively large margin. BoT only lags behind the CSV [1] on the MATH dataset. Yet this is not a fair comparison, since CSV additionally heavily relies on the GPT-4 code interpreter, whereas BoT achieves a competitive performance as an automated prompting framework without relying on any tools such as the GPT-4 code interpreter. Under a fair comparison with other related works on the MATH dataset, as shown in Section F of the appendix, BoT consistently achieves the highest problem solving rate on different sub-categories. Furthermore, for the Game of 24, according to the results shown in Table 2, BoT is $9.7\%$ higher than the current best method ToT [2]. Besides, what is more important is that BoT is an automated prompting framework that generates effective reasoning chains by collecting trial-and-error reasoning experiences without introducing human annotations.
>
>
> [1]. hou, Aojun, et al. Solving Challenging Math Word Problems Using GPT-4 Code Interpreter with Code-based Self-Verification, Arxiv 2023.
>
> [2]. Yao, Shunyu, et al. Tree of Thoughts: Deliberate Problem Solving with Large Language Models, Arxiv 2023.

---

### Official Review · Reviewer_dFck · 2023-11-05

**Soundness:** 3 good
**Presentation:** 2 fair
**Contribution:** 2 fair
**Rating:** 6
**Confidence:** 3

**Summary:**

The paper presents an extension of the Chain of Thought (CoT) and Tree of Thoughts (ToT) method, named Boosting of Thoughts (BoT). BoT refines the problem-solving process in large language models (LLMs). BoT harnesses error analysis to improve the LLM's problem-solving accuracy iteratively. The "Boosting of Thoughts" (BoT) procedure is a two-step process that first generates a diversity of reasoning paths from a Large Language Model (LLM) in the form of a weighted binary tree, enhancing problem-solving by creating a hierarchy of potential solutions. Then, it employs a novel aggregation strategy that iteratively refines and combines these paths. Through best-first and greedy aggregations, BoT selects and optimizes the most promising chain of thought, using iterative feedback to progressively improve the LLM's performance on complex problem-solving tasks. The paper reports improved performance on complex mathematical problems when tested with GPT-4 and LLAMA2, compared to CoT and ToT.

**Strengths:**

1. This is an innovative extension of the Chain-of-Thought (CoT) and Tree-of-Thought (ToT) methods. Compared to CoT and ToT, the author adopts the idea on leveraging error analysis to refine the LLM. This can be a limitation of CoT and ToT, as they do not conduct error analysis and more importantly, learn from errors. The motivation is intuitive and clear.

2. Unlike ToT, which expands multiple reasoning tree branches, the BoT method iteratively refines a single line of thought. This focus on iteration rather than expansion allows for a more concentrated and efficient improvement of the reasoning path. The computation moves from exploring the tree into learning from erroneous trials.

3. The Boosting of Thoughts (BoT) concept shows a clear advancement in problem-solving methodologies within large language models. It effectively combines generation and evaluation steps to progressively enhance reasoning, demonstrating a significant leap in the model's ability to handle complex tasks.

4. The experiments are clear, the results are effective. And all experiments are classic experiments from CoT and ToT, so it is clear to compare BoT’s performance over CoT and ToT.

**Weaknesses:**

The mauscript need polished in their figures' presentation, e.g., the authors need give more detailed examples in Fig1.

**Questions:**

Q1: In the prompt, I wonder whether the “error input” are included, or only the “experience” is included? From figure 1, I only see “error report” like “step 1 is not closer to 24”, no “error input” like what is step 1, 2, 3. How the LLM know what step 1 mean, and how can LLM learn from error, if LLM does not know specific input?
Q2: How about you consider the entire (input, error analysis) as an In-context Learning example? Then the entire method is similar to CoT, meaning that you can manually construct an exemplar consisting of (input, error analysis) pair. Then use the CoT idea to follow the strategy to generate analysis and think about the correct answer.

---

> ### Author Response · Authors · 2023-11-22
> **Reply to Reviewer dFck**
>
> Thanks for your comments and we reply to them below.
>
> __Response to W1&Q1__:
>
> We have revised and polished all Figures, especially Figure 1, in the submission. To provide further clarity, additional details about each component of BoT, as illustrated in Figure 1 and Figure 2, have been included in sections A, C, and D of the appendix. In particular, we have supplemented the appendix with additional experimental results, including the experience in Table 7 and Table 8, generated by BoT. Furthermore, we have enhanced the organization of the source code within the _examples/BoostingOfThought/_ directory to provide a more comprehensive overview of the BoT implementation. For instance, within the code file _BoT\_commenter.py_, one can find the process for generating experience using LLMs, while _BoT\_reasoner.py_ details how this experience is structured and incorporated into the prompt for subsequent iterations of reasoning.
>
> Specifically, both the 'error input' and 'experience' are included in the prompt, but the 'error input' is embraced as the sub-block of 'experience'. In fact, the experience comprises input reasoning steps, the conclusion, error analysis, advice, and the confidence score, all of which are generated automatically by LLMs. As each iteration of BoT produces one such experience for the reasoning chain, LLMs can gradually generate the right answer due to the experience accumulation in the prompt over iterations. Tables 7 and 8 of the appendix provide a direct example of how experience is organized in the prompt.
>
> The confusion arising from Figure 1 primarily stems from our intention to first emphasize the significance and effectiveness of an ensemble of trial-and-error reasoning experiences within the context of the proposed automated prompting framework BoT. To mitigate the confusion, we have incorporated additional explanations into Figure 1.
>
> __Response to Q2__:
>
> Your further suggestion to include manually constructed (input, error analysis) as an in-context learning example in the prompt, similar to the CoT approach, is highly insightful. We also contemplated this intriguing idea but had to abandon it due to three key concerns. First, relying on human priors to manually design error analysis and advice is time-consuming, inefficient, and hard to generalize well to different tasks. Second, BoT, an automated prompting framework with experience generated by LLMs, is able to achieve a competitive or even top problem-solving rate on multiple mathematical problems, as shown by experimental results. Third, embracing human knowledge in the prompt may have limited applications in real-world usage due to possible bias, misleading, and security issues brought by others.

---

### Official Review · Reviewer_x4s4 · 2023-11-07

**Soundness:** 2 fair
**Presentation:** 3 good
**Contribution:** 2 fair
**Rating:** 5
**Confidence:** 3

**Summary:**

The paper proposes a new framework Boosting of Thoughts (BoT) with large language models (LLMs) for task-specific prompting. It provides how to construct prompts and use the trial-and-error reasoning approach to interact with the LLM to generate the final responses. The experiments show the effectiveness of the proposed method.

**Strengths:**

- Prompt engineering is a non-trivial task, and crafting effective prompts may require specialized training for human experts. The paper introduces an innovative framework for iterative prompting, leveraging LLM's feedback on its own reasoning, thereby reducing the need for human prompt engineering.

- Addressing complex problems is crucial in LLM applications. This approach effectively demonstrates the power of prompt engineering and expands the capabilities of LLMs without the need for retraining or fine-tuning. Experiments conducted on multiple datasets show competitive performance compared to other prompting approaches.

**Weaknesses:**

- I agree that prompt engineering is crucial for LLM applications. However, it's worth noting that prompt engineering is often model-dependent, and the techniques may evolve as LLM capabilities improve. This may not offer long-term guidance for research unless it uncovers fundamental insights. This distinction is critical in differentiating academic research from practical production. Therefore, while the paper does offer valuable techniques for prompting the model and achieving good results on evaluation sets, it lacks in-depth discussion of the underlying reasons. This makes the paper better suited for application-oriented conferences rather than ICLR.

- LLMs can be unstable and prone to hallucination, which could result in bad or incorrect feedback when using the Boosting of Thoughts (BoT) iterative prompting framework. Is there analysis on the impact of "bad" LLM feedback? Further, as the iterative produces are automatic, spurious feedback could get amplified over iterations. Some discuss may be necessary.

-  Details are lacking on key components like aggregation strategies and generating edge weights for trees. More analysis or ablation studies are also helpful.

**Questions:**

- In Section 3.2, it is not quite clear how to calculate the weights for the weighted binary tree.

---

> ### Author Response · Authors · 2023-11-22
> **Reply to Reviewer x4s4 - Part 1**
>
> __Response to W1__:
>
> First, we cannot agree with the reviewer that this work does not offer long-term guidance for research and does not uncover fundamental insights. We do have offered fundamental insights for reasoning with LLMs. Traditionally, LLMs cannot solve complex mathematical problems sufficiently well. We show that there is a way to generate and select thoughts, and leverage the error analysis generated by LLM itself for these thoughts and feeding such error report back to prompt the LLM, we can progressively and accumulatively derive the right prompt required for LLM to generate the solution to complex mathematical problems. It shows the latent capability and potential of LLM can be unleashed through proper triggering from the error analysis and advice on the generated and selected reasoning steps. This process is fundamentally similar to humans who possess basic mathematical skills and are tasked to solve a math question. For solving a particular question, one sometimes must learn from not only demonstrations but also prior trials and errors to progressively discover the reasoning path. Intuitively speaking, Chain-of-Thoughts (CoT) [1] shows that demonstration via examples is conducive to problem solving. However, we point out that demonstration is not enough and also hard to obtain anyway. One must also practice and encounter errors and absorb such trial-and-errors into his/her knowledge base to solve the problem. So does an LLM. Thus, BoT offers fundamental insights and guidance on how to enable LLMs to generate effective reasoning steps for complex problem solving, essentially through retrospective error analysis, without requiring human demonstrations.
>
> With these insights, the research of prompt engineering on inducing the reasoning capability of LLMs can focus on how to generate a compiled list of trial-and-error history (error analysis report from solving the problem) instead of introducing more human priors (examples) to the prompt like existing CoT, ToT methods would do, which we believe is both insufficient and hard to acquire. This, in turn, makes BoT an automated prompting framework because by iteratively collecting effective error analysis in the prompt without human annotations, LLMs can be guided to produce a correct reasoning chain toward problem solving.
>
> Second, our insight is NOT model dependent---the same method can be successfully applied to several LLMs, such as gpt-3.5-turbo, gpt-4, and Llama2, according to our experiments. In particular, we utilize the gpt-3.5-turbo model to generate examples shown in the Appendix. The results still support our insights and conclusion. In Section B of the Appendix, we present more discussions on the fundamental insights of BoT. Furthermore, the base model in BoT's boosting mechanism does not have to be specific tree thought structures, such as ToT [2]. We chose ToT here for its effectiveness and simplicity. Therefore, the fundamental concept underpinning the proposed BoT's outstanding performance with an automated prompting framework lies in recognizing the fundamental cycle of thought generation, selection and error analysis to be fed back into LLM prompting as advice and guidance, which unlocks LLM's ability to solve complex problems. We propose BoT as a general trial-and-error prompting framework to enhance the reasoning capabilities of LLMs (which proves to be effective on multiple LLMs in general), rather than a model-dependent engineering trick that will fade away as models evolve.
>
> Third, we are certainly not targeting applications or practical production related to LLMs. While most real-world LLM applications are regarding chatting with humans (i.e., ChatGPT), it is an emerging research question arising in the NeurIPS/ICLR/ICML/AAAI community to ask whether LLMs can solve complex mathematical questions (which is certainly not applications today, but simply a research question asked by the research community out of pure intellectual interests). For example, Chain of Thoughts (CoT) arises in NeurIPS 2022. To answer this fundamental question, we show the possibility of prompting LLM in a different way that no other work on Chain of Thoughts has tried out before, i.e., through a log of trial-and-error analysis, and show that this style of prompting (instead of just demonstrating human priors to LLMs) is key to unlocking LLMs' capabilities to solve complex mathematical problems. The underlying reason is that simply demonstrating to LLMs the steps to solve a math problem is not enough. We must also leverage a compiled list of trials and errors obtained from attempting to solve the problem so that the LLM progressively learns to avoid pitfalls, just like humans. Our experiments have substantiated our claim by showing that without human priors, BoT can achieve outstanding performance at solving various mathematical problems via the proposed thought generation, aggregation, and error analysis accumulated to be used for prompting LLMs.

---

> ### Author Response · Authors · 2023-11-22
> **Reply to Reviewer x4s4 - Part 2**
>
> __Response to W2__:
>
> In Section E of the appendix, we have discussed that the spurious feedback may lead the LLMs to generate reasoning steps that are logically incorrect or do not adhere to any of the task rules. Table 7 of the Appendix presents the corresponding results of BoT when the spurious feedback is included as the experience in the prompt. However, we argue that spurious feedback will NOT be amplified over iterations; instead, thanks to the iterative mechanism of BoT, its negative impact on the generated reasoning steps can be mitigated or even entirely rectified in subsequent iterations. Specifically, as the wrong reasoning steps caused by spurious feedback contain obvious mistakes, it is likely that LLMs tend to generate correct error analysis by comparing to the final target at some later steps and provide effective suggestions for revisions. With this new experience included in the prompt, BoT is capable of generating correct thoughts (reasoning steps). This is further mitigated by the proposed thought generation process through an ensemble of binary trees and aggregating them to acquire the better chain of thoughts to be used in error analysis in each iteration.
> As demonstrated by the experience in Table 8, BoT produces detailed error reports and revision suggestions, resulting in a rational thought generation process illustrated in Table 7 of the Appendix.
>
> The advantage of BoT, which leverages iterations to mitigate the detrimental effects of invalid or erroneous feedback (i.e., the LLMs progressively avoid pitfalls during the BoT process), is evident in Figure 4. Notably, the performance of BoT exhibits consistent enhancement as the number of iterations increases. This implies both the significance of accumulating trial-and-error experiences iteratively and the capacity of subsequent experiences to rectify (avoid) errors in earlier experiences.
>
> __Answer to W3 and Q1__:
>
> We have added more details and discussions on edge weight computation for trees and the aggregation strategies in Sections C and D of the Appendix. In summary, to calculate the edge weight $V_{i-1, i}$ between two reasoning steps, represented as nodes in the tree and denoted by $z_{i-1}$ and $z_{i}$, LLMs are utilized to evaluate the entire reasoning chain $z_{1, ..., i-1, i}$, which forms a branch of the tree with $z_1$ acting as the root node. The corresponding prompt used by LLMs for this weight computation can be found in Section A of the Appendix and the source code _examples/BoostingOfThought/BoT\_reasoner.py_.
>
> [1]. Wei, Jason, et al., Chain-of-thought prompting elicits reasoning in large language models, Advances in Neural Information Processing Systems 35 (2022): 24824-24837.
>
> [2]. Yao, Shunyu, et al., Tree of Thoughts: Deliberate Problem Solving with Large Language Models, Arxiv 2023.
>
> [3]. Besta, Maciej, et al., Graph of thoughts: Solving elaborate problems with large language models, Arxiv 2023.
>
> We endeavour to address reviewer questions and concerns and improve the quality of this research. It is much appreciated if the reviewer could reassess the value of our contribution after reading the rebuttal.

---

> > ### Comment · Reviewer_x4s4 · 2023-12-04
> >
> > For responses of W2, it is questionable whether an LLM can consistantly identify its own errors during iterations. It seems likely that, due to their limited capabilities, LLMs might become entrapped in their own incorrect reasoning. For instance, when querying several LLMs, including ChatGPT and Claude, with a very simple question, "If A>B and B<C, then what is the relationship between A and C?", I found that except for GPT-4, all others struggled with this query. Also, I followed to the methodology described in this paper, but still found that LLMs consistently assigned a score of 1.0 to incorrect answers. However, if I modified the prompt to "If A>B and C>B,...", it leads most LLMs to provide correct answers without any prompt engineering. This observation suggests that LLMs may only learn language patterns rather than possessing true reasoning capabilities. Therefore, while I agree that BoT iterations can enhance performance, it is better for the authors to acknowledge that under certain conditions, the inherent limitations of LLMs might lead to failure.
> >
> > Responses of W3 are accepted.

---

> ### Comment · Reviewer_x4s4 · 2023-12-04
>
> I am certain that prompt engineering is cruicial, particularly in practical applications. However, I still have some doubts about its fundamental importance for AGI. While human communication needs sophisticated and nuanced skills, we do not always employ complex skills to construct language when seeking straightforward answers, which is what we desire from LLMs. In the future, AGI should be able to respond to humans without the need for complicated prompt engineering. But, of course, this is my personal opinion, and I may not use it as the standard for evaluating the paper. Therefore, I am open to accepting the authors' response on this matter.

---

### Meta-Review · Area_Chair_WrDu · 2023-12-05

**Metareview:**

The paper introduces "Boosting of Thoughts" (BoT), a novel approach for problem-solving in Large Language Models (LLMs), marked by its conceptually clear framework that utilizes an iterative trial-and-error mechanism for prompt refinement. The methodology stands out for its originality, significantly advancing over existing methods like Chain of Thought (CoT) and Tree of Thoughts (ToT) in mathematical datasets.
The authors have addressed most of the reviewer's concerns during the rebuttal phase, enhancing the paper's clarity and depth. However, the paper does have limitations, including the lack of explicit supervised evaluation of the model-generated error assessments and scores. Additionally, the BoT methodology might incur high costs in API tokens due to its dependence on multiple iterations and frequent error analysis, an aspect that warrants further exploration to ensure fair comparison with CoT and ToT. Overall, considering the strengths and addressing the limitations, I recommend accepting this paper for ICLR.

**Justification For Why Not Higher Score:**

the BoT methodology might incur high costs in API tokens due to its dependence on multiple iterations and frequent error analysis, an aspect that warrants further exploration to ensure fair comparison with CoT and ToT.

**Justification For Why Not Lower Score:**

The approach looks novel and may inspire follow-up research on this.

---

### Decision · Program_Chairs · 2024-01-16

Accept (poster)